

# Nutrient cycling in the Baltic Sea - results from a 30-year physical-biogeochemical reanalysis

Ye Liu[1], H.E. Markus Meier[2,1], Kari Eolila[1]

[1]Swedish Meteorological and Hydrological Institute, Norrköping, Sweden

[2]Leibniz Institute for Baltic Sea Research Warnemünde, Rostock, Germany

*Correspondence to:* Ye Liu (ye.liu@smhi.se)

**Abstract.** The long-term oxygen and nutrient cycles in the Baltic Sea are reconstructed using the Swedish Coastal and Ocean Biogeochemical model (SCOBI) coupled to the Rossby Centre Ocean model (RCO). Two simulations covering the period 1970–1999 are carried out with and without data assimilation, respectively. Here,
the "weakly coupled" scheme with the Ensemble Optimal Interpolation (EnOI) method is adopted to assimilate the observed profiles in the reanalysis system. The simulation results show considerable improvements in both oxygen and nutrient concentrations in the reanalysis relative to the free run. Further, the results suggest that the assimilation of biogeochemical observations has a significant effect on the simulation of the oxygen dependent dynamics of biogeochemical cycles. From the reanalysis, nutrient transports between subbasins, between the
coastal zone and the open sea, and across latitudinal and longitudinal cross sections, are calculated. Further, bottom areas of nutrient import or export are examined. Our results emphasize the important role of the Baltic proper for the entire Baltic Sea, with large net exports of nutrients into the surrounding subbasins (except the phosphorus transport into the Gulf of Riga and the nitrogen transports into the Gulf of Riga and Danish Straits). In agreement with previous studies, we found that the Bothnian Sea imports large amounts of phosphorus from
the Baltic proper that are buried in this subbasin. For the calculation of subbasin budgets, it is crucial where the lateral borders of the subbasins are located, because net transports may change sign with the location of the border. Although the overall transport patterns resemble the results of previous studies, our calculated estimates differ in detail considerably.





**Keywords**: reanalysis; data assimilation; numerical modelling; Baltic Sea; biogeochemical cycles; nutrient
budgets

# 1 Introduction

The water exchange between the Baltic Sea and the North Sea is restricted by the narrows and sills in the Danish
transition zone (Fig. 1). The hydrography of the Baltic Sea also depends on freshwater from rivers, which causes
large salinity gradients between the surface layer and the saltier bottom layer, and between the northern subbasins
and the entrance area (e.g. Meier and Kauker, 2003). The low-saline outflowing surface water is separated from
high-saline inflowing bottom water by a transition layer, the halocline. Furthermore, the bottom water in the deep
subbasins is ventilated mainly by so-called Major Baltic saltwater Inflows (MBIs) (Matthäus and Franck, 1992;
Fischer and Matthäus, 1996). MBIs can significantly affect the living conditions in the deep basins because of the
inflow of large volumes of oxygen-rich water into the Baltic Sea (e.g. Fu, 2013). In the Baltic Sea, the density
stratification and long water residence time hamper the ventilation of deep waters. As a result, oxygen deficiency
is a common feature. Additionally, nutrient loads from agriculture and other activities of the large population in
the catchment area increased nutrient concentrations in the water column. Actually, eutrophication has become a
large environmental problem in the Baltic Sea in recent decades (e.g. Boesch et al., 2008; Pawlak et al., 2009;
Wulff et al., 2001; Andersen et al., 2015). Therefore, accurate estimates of the ecological state and nutrient and
water exchange between subbasins and between the coastal zone and the open sea are of particular importance in
managing the marine environment system. Moreover, it is helpful to understand some primary biogeochemical
dynamic processes in the Baltic Sea.

On one hand, the estimation of biogeochemical processes, ecological state and nutrient exchange may rely on
coupled marine ecosystem-circulation models (e.g. Neumann et al., 2002; Eilola et al., 2009; 2011). However,
addressing biogeochemical cycles is a challenging task due to the complexity of the system. Obviously, there are
large uncertainties in marine ecological simulations (e.g. Meier et al., 2012). In contrast to the modelling of the
physics of the atmosphere or ocean, where a basic description of the motion is provided by conservation
equations, there is no basic set of equations that describe the marine ecosystem. Many biogeochemical processes





are still poorly known and their uncertainties are difficult to quantify accurately. In general, coupled physical-
biogeochemical models use a variety of biological formulations (either empirical or mechanistic) to update
biogeochemical concentrations. As a result, the model formulation and the reliability of their parameterizations
play a key role in determining the simulation accuracy of biogeochemical processes. In reality these processes
governing the interactions between biogeochemical compartments vary in space and time (Losa et al., 2004;
Doney, 1999). These potential sources of errors limit the applicability of the models, unless the parameters can be
better determined. Further, the imperfect initial conditions and forcing also cause uncertainty of the simulation
results.

On the other hand, estimating nutrient budgets and transports between subbasins may directly rely on
observations (Savchuk, 2005). The estimation accuracy depends on the spatial and temporal coverage of the
measurements. It seems unrealistic to obtain synoptic descriptions of vast ocean basins using direct observations
because of the temporal and spatial patchiness of biological parameters (Garcia et al., 2010). Although the data
coverage in the Baltic Sea has gradually increased over time, the lack of observations still makes it difficult to
estimate reliable biogeochemical cycles. Today, the availability of satellite sensor data like ocean color data from
the OCTS (Ocean Color and Temperature Sensor) and from the SeaWiFS has provided the best spatial coverage
of measurements. However, these sensors only give an estimate of a few biogeochemical parameters at the
surface of the marine ecosystem, and not the state of the entire marine ecosystem. Continuous deep observations
are only possible with in situ sensors, which have been deployed at only a limited number of stations (Claustre et
al., 2010).

Given the coverage of observations and model deficiency, high-resolution reanalysis data may be considered as
an advisable way to achieve reliable estimates of the physical-biogeochemical state. Data assimilation has been
developed for the synthesis between models and observations. However, the assimilation of data into the coupled
physical-biogeochemical model is confronted by various theoretical and practical challenges. For example, the
response of the three-dimensional biogeochemical model to external forcing caused by the physical model is
highly non-linear. Further, it is difficult to use the biological observational information to control the effect of
errors in the ocean physics that also cascade onto the biogeochemistry (Beal et al., 2010). Presently, the use of
data assimilation to complement ecosystem modeling efforts has gained widespread attention (e.g. Hoteit et al.,
2003; Allen et al., 2003; Natvik and Evensen, 2003; Hoteit et al., 2005; Triantafyllou et al., 2007; While et al.,
2012; Triantafyllou et al., 2013). A comprehensive review of biological data assimilation experiments can be
found in Gregg et al. (2009). In the Baltic Sea, the biogeochemical data assimilation has started to become a



research focus. For example, Liu et al. (2014) used the Ensemble Optimal Interpolation (EnOI) method to

improve the multi-annual, high-resolution modelling of biogeochemical cycles in the Baltic Sea.

However, in Liu et al. (2014), only a shorter assimilation experiment for a 10-year period is presented, and so far the stability of the assimilation scheme in multi-decadal simulations has not been shown. Fu (2016) analyzed the response of the coupled physical-biogeochemical model to the improved hydrodynamics in the Baltic Sea. Recently several data assimilation studies have focused on the historical reanalysis of salinity and temperature in

the Baltic Sea (e.g. Fu et al., 2012; Liu et al., 2013; 2014). Reanalysis has helped enormously in making the historical record of observed ocean parameters more homogeneous and useful for many purposes. For instance, ocean reanalysis data have been applied in research on ocean climate variability as well as on the variability of biochemistry and ecosystems (e.g. Bengtsson et al., 2004; Carton et al., 2005; Friedrichs et al., 2006). Ocean reanalysis can also be used for the validation of a wide range of model results (e.g. Fu et al., 2011; Fontana et al.,

2013). Moreover, reanalysis in the ocean is beneficial to the identification and correction of deficiencies in the observational records, as well as filling the gaps in observations.

The reanalysis is mainly based on a reliable model, an effective observational system, and a good assimilation method. The Baltic coastal shelf observation systems have been largely improved by the joint efforts of the countries surrounding the Baltic Sea. For example, the International Council for the Exploration of the Sea

(ICES) (http://www.ices.dk) and the Swedish Oceanographic Data Centre (SHARK) (http://sharkweb.smhi.se) are monitoring the long time scale of the Baltic Sea. Furthermore, the Baltic Sea Operational Oceanographic System (BOOS) (http://www.boos.org/) is providing near real-time observations. Moreover, the publicly available databases of the Baltic Nest Institute (BNI) (http://www.balticnest.org) store the data from the partner institutes (see http://nest.su.se/bed/hydro_chem.shtml). As a result, a comprehensive data set is collected for the

Baltic Sea region. In this study, observations since 1970 are used, as the data coverage since the 1970s has been satisfactory. These observations provide an important tool for validating the model results and identifying existing problems of the ecosystem model. Furthermore, ecological and biogeochemical Baltic Sea models with varying levels of complexity are under active development (Eilola et al., 2009; 2011; Almroth-Rosell et al., 2011; 2015; Neumann et al., 2002; Edvelvang et al., 2005; Ryabchenko et al., 2016; Savchuk et al., 2008), and recently

coupled high-resolution three-dimensional (3D) models, including for the North Sea, have been developed as well (Maar et al., 2011; Daewel and Schrum, 2013).

This study presents the first attempt to focus on a 30-year (1970–1999) biogeochemical reanalysis to reconstruct multi-decadal variations in the biological state of the Baltic Sea. The present paper focuses on the



assimilation of the profiles of temperature, salinity, nutrients and oxygen in the Baltic Sea following Liu et al.

(2014). The aim is a better assessment of historical changes in the nutrient budgets of the water column and

sediments, as well as of the nutrient exchanges between subbasins and between the coastal zone and the open sea.

This paper is organized as follows. The physical and biogeochemical models are described in Section 2. Then

the method of the reanalysis study and the observational data are introduced in Sections 3 and 4, respectively. The

experiment results, including comparisons with observations, are presented in Section 5. Finally, in Section 6,

discussions and conclusions finalize the paper.

**2 Models**

The RCO (Rossby Centre Ocean) model is a Bryan–Cox–Semtner primitive equation circulation model with a

free surface (Killworth et al., 1991). Its open boundary conditions are implemented in the northern Kattegat,

based on the prescribed sea surface heights at the lateral boundary (Stevens, 1991). An Orlanski radiation

condition (Orlanski, 1976) is used to address the case of outflow, and the temperature and salinity variables are

nudged toward climatologically annual mean profiles to deal with inflows (Meier et al., 2003). A Hibler-type

dynamic–thermodynamic sea ice model (Hibler, 1979) with elastic–viscous–plastic rheology (Hunke and

Dukowicz, 1997) and a two-equation turbulence closure scheme of the $k$–$\varepsilon$ type with flux boundary conditions

(Meier, 2001) have been embedded into RCO. The deep-water mixing is assumed to be inversely proportional to

the Brunt–Väisälä frequency, with the proportionality factor based on dissipation measurements in the Eastern

Gotland Basin (Lass et al., 2003). In its present version, RCO is used with a horizontal resolution of 2 nautical

miles (3.7 km) and 83 vertical levels, with layer thicknesses of 3 m. RCO allows direct communication between

bottom boxes of the step-like topography (Beckmann and Döscher, 1997). A flux-corrected, monotonicity-

preserving transport (FCT) scheme is applied in RCO (Gerdes et al., 1991). RCO has no explicit horizontal

diffusion. For further details of the model setup, the reader is referred to Meier et al. (2003) and Meier (2007).

The biogeochemical model called SCOBI (Swedish Coastal and Ocean Biogeochemical model) has been

developed to study the biogeochemical nutrient cycling in the sea (Marmefeldt et al., 1999; Eilola et al., 2009;

Almroth-Rosell et al., 2011; 2015). This model handles biological and ecological processes in the sea as well as

sediment nutrient dynamics. SCOBI has been coupled to RCO (e.g. Eilola et al., 2012; Eilola et al., 2013; Eilola

et al., 2014). With the help of a simplified wave model, resuspension of organic matter is calculated from the

wave and current-induced shear stresses (Almroth-Rosell et al., 2011). SCOBI has a constant carbon (C) to





chlorophyll (Chl) ratio C:Chl = 50 (mg C (mg Chl)$^{-1}$), and the production of phytoplankton assimilates carbon (C), nitrogen (N) and phosphorus (P) according to the Redfield molar ratio (C:N:P = 106:16:1) (Eilola et al., 2009). The molar ratio of a complete oxidation of the remineralized nutrients is $O_2$:C = 138. For further details of

the SCOBI model, the reader is referred to Eilola et al. (2009, 2011) and Almroth-Rosell et al. (2011).

The model (RCO-SCOBI) is forced by atmospheric forcing data calculated from regionalized ERA-40 data using a regional atmosphere model RCA (Samuelsson et al., 2011). The horizontal resolution of RCA is 25 km. A bias correction method following Meier et al. (2011) is applied to the wind speed. Monthly mean river runoff observations (Bergström and Carlsson, 1994) are used for the hydrological forcing. Monthly nutrient loads are

calculated from historical data (Savchuk et al., 2012).

**3 The Dataset**

The assimilated observations in this study are both physical (temperature and salinity) and biogeochemical variables (oxygen, nitrate, phosphate and ammonium) from the Swedish Oceanographic Data Centre's SHARK database. Before assimilation, the data are quality controlled. These controls include checks of locations and

duplication, and examination of differences between forecasts and observations. A profile was eliminated from the assimilation procedure when the station was located on land defined by the RCO bathymetry. We also removed observations when the difference between model forecasting field and observations exceeded the given standard maximum deviation (for example 4.0 mL L$^{-1}$ for oxygen concentration). We used an average of the observations in the same layer when there was more than one observation per layer. These observations cover

almost the whole Baltic Sea including Kattegat and the Danish Straits. Fig. 2 shows the number of biogeochemical observations in different subbasins, and the temporal distribution of these biogeochemical observations. The number of observations is inhomogeneous in both temporal and spatial distribution over the period from 1970 to 1999. There are relatively more observations in the Baltic proper than in other subbasins. In the Gulf of Riga, a minimum number of observation profiles (30 for oxygen, 30 for phosphate, 28 for nitrate and

28 for ammonium) are found. Obviously, the number of observations during the period of 1988-1994 is greater than that during other periods. Further, there are generally less observations from 1981-1983 than during other periods. The maximum number of observation profiles occurred in 1991 for oxygen (1,844), phosphate (1,728) and nitrate (1,758). However, the number of ammonium observation profiles has a maximum value of 1,222 in 1992. Moreover, the number of the oxygen and ammonium observations is largest and smallest, respectively,



compared to the other variables. These observations from SHARK and the Baltic Environmental Database (BED) at the Baltic Nest Institute (http://nest.su.se/bed) are used both to validate the model results and for the assimilation.

## 4 Methodology and Experimental Setup

Here we briefly describe the configuration of the data assimilation system of this study. We focus on the state
estimation via EnOI. The distribution of stochastic errors are assumed to be Gaussian and non-biased. A total 88 model samples by "running selection" are adopted to obtain a quasi-stationary background error covariance (BEC) (Liu et al., 2013). An adaptive scaling factor was calculated to adapt to the instantaneous forecast error variance before each local analysis (Liu et al., 2013; 2014).

Based on the above configuration, two experiments from January 1970 to December 1999 have been carried
out. One experiment is a simulation without data assimilation (FREE). The other simulation is constrained by observations using the "weakly coupled" assimilation scheme based upon the EnOI method following from Liu et al. (2014) (REANA). Both simulations, FREE and REANA, are initialized for January 1970, taken from an earlier run with RCO-SCOBI. For details of the method, the reader is referred to Liu et al. (2014). However, in Liu et al. (2014), only a shorter assimilation experiment for a 10-year period is presented, and so far the stability
of the assimilation scheme in multi-decadal simulations has not been shown. Following Liu et al. (2014), our REANA experiment assimilated both physical and biogeochemical observations. In this study, we focus mainly on nutrient budgets and transports since changes of physical parameters are similar to those in Liu et al. (2013).

## 5 Results

In the following sections, we evaluate the impact of data assimilation on the long-term evolution of biases
(Section 5.1), vertical (Section 5.2) and horizontal (Section 5.3) distributions of nutrient concentrations, and long-term trends in eutrophication (excess of nutrients in the water column) as indicated by Secchi depth (Section 5.4). For the evaluation of time series of simulated oxygen, nitrate, ammonium and phosphate concentrations, the reader is referred to Liu et al. (2014, their Fig. 6 and 7). After the evaluation of the assimilation method, we focus on the analysis of the nutrient cycling in the Baltic Sea based upon our reanalysis data that we consider to be the
best available data set for such an analysis. In particular, we analyze the horizontal circulation of nutrients





(Section 5.5), the horizontal distribution of nutrient sources and sinks, the nutrient exchange between the coastal zone and the open sea (Section 5.6), and the nutrient budgets of subbasins (Section 5.7).

## 5.1 Temporal evolution of biases

To assess the results with (REANA) and without (FREE) data assimilation, the overall monthly mean RMSDs (root mean square differences) of oxygen, nitrate, phosphate and ammonium were calculated relative to observations during the whole integration period (Fig. 3). Here it should be noted that the RMSDs were calculated before the time of assimilation analysis, and the corresponding observations were not yet assimilated into RCO-SCOBI (Liu et al., 2014). The data assimilation has significantly positive impact on the model simulation. Generally, the RMSDs of the oxygen and nutrient concentrations in REANA are smaller than that of FREE. However, the improvements of these four variables have different variation characteristics caused by the assimilating of biogeochemical observations. The RMSD of oxygen is mostly greater and smaller than 1.0 mL L$^{-1}$ for the FREE and REANA, respectively. The mean RMSD of oxygen during this period has been reduced by 59% (from 1.43 to 0.59 mL L$^{-1}$). Similar improvements also appear in nitrate and phosphate concentrations. The RMSDs of nitrate and phosphate in REANA were reduced by 46% (from 2.04 to 1.11 mmol m$^{-3}$) and 78% (from 1.05 to 0.23 mmol m$^{-3}$) relative to that in FREE, respectively. Furthermore, the variability of RMSD of phosphate in FREE is large during the first 10 years, and decreases afterwards with time. However, the data assimilation cannot always improve the model results (Liu et al., 2014). For instance, although the overall RMSD of ammonium is reduced by 45% (from 1.15 to 0.63 mmol m$^{-3}$), the ammonium concentrations in REANA become worse relative to those in FREE during some months. An example appears in February 1975 when the RMSD of the ammonium concentrations in REANA (3.07 mmol m$^{-3}$) is greater than that in FREE (2.75 mmol m$^{-3}$). These results are similar to the findings by Liu et al. (2014). However, here we show that the 30-year-long assimilation is stable, and that the RMSD of phosphate concentrations decreases even further with data assimilation after 10 years.

## 5.2 The seasonal cycle of nutrients

The long-term average seasonal cycles of temperature and inorganic nutrients at monitoring station BY15 at Gotland Deep (for the location, see Fig. 1) give a hint of how nutrient dynamics in the Baltic proper work with





and without data assimilation (Fig. 4). The surface layer temperature and stratification showed rapid increase in April to May, with concurrent rapid decrease of nutrient concentrations due to primary production down to 50-60 m depths. The cooling and increased vertical mixing in the autumn and winter reduced temperatures and brought

nutrients from the deeper layers into the surface layers. RCO-SCOBI has captured these variations. However, compared to BED, the model has obvious biases, such as from late winter to early spring temperature stratification in FREE around the 30-50m depth, higher concentration of nutrients at the 50-60m depth, stronger vertical stratification of nutrient concentrations and less decrease of nutrients in the summer, especially below the thermocline, as well as also in the surface layers for phosphate. One reason for the biases is due to the vertical

displacement of the halocline that is too shallow in RCO (e.g. Fig. 4 in Liu et al., 2014). The causes for the model bias in nutrient depletion below the summer thermocline are not known, but possible reasons are discussed by Eilola et al. (2011). The reanalysis has significantly reduced all these biases, as well as an improved model description of vertical transports of nutrients in the layers above the halocline.

**5.3 Spatial variations of late winter nutrient concentrations**

The average March concentrations of dissolved inorganic phosphorus (DIP) and nitrogen (DIN) in the upper layers (0-10m), as well as their ratio (DIN:DIP), were calculated (Fig. 5). In BED the highest concentration of DIP occurs in the Gulf of Riga and the Gulf of Finland. Relatively high concentrations of DIP are also found along the western coast of the Gotland Basin. The DIP concentrations in the Bothnian Sea and Bothnian Bay are obviously lower than in other regions. Generally, the DIP in FREE has been largely overestimated in all regions

relative to BED, especially in the Gotland Basin and Bornholm Basin. The low DIP concentrations in BED are appearing at the eastern coast of the Eastern Gotland Basin. In FREE, this spatial feature of DIP concentrations is not found. Further, high concentrations of DIN in BED occur in coastal waters close to the river mouths of the major rivers in the southern Baltic proper. DIN concentrations in the Gulf of Finland and in the Gulf of Riga are also high, and cover large areas of these gulfs. Unlike the BED data, the DIN in FREE also has high

concentrations in the entire southern and eastern coastal zones of the Baltic proper. As a result, FREE shows a gradient in DIN concentrations between the coastal zone and the open sea in the entire southern Baltic proper. The DIN and DIP patterns result in high and low DIN:DIP ratios in the Bothnian Bay and Baltic proper, respectively. The highest DIN:DIP ratios are found in the Bothnian Bay in BED and in the Gulf of Riga in FREE. RCO-SCOBI has captured this large-scale pattern, but there are substantial regional differences. By the




constraints of the observation information, REANA has improved the spatial distributions of DIN and DIP significantly. In particular, DIP concentrations in REANA are much closer to observations.

## 5.4 Trends in Secchi depth

An important indicator of water quality is water transparency, as it directly affects primary production by light attenuation. Secchi depth is a measure of water transparency. In general, the reanalysis increased the Secchi depth

relative to FREE (Fig. 6). This change is explained by the decrease of dissolved nutrient concentrations in REANA relative to FREE, with consequently reduced concentrations of organic matter (not shown). Further, the enhanced stratification limits the entrainment of nutrients from below into the euphotic zone, and thus decreases the nutrient concentrations in the upper and mid layers (Fig. 4). The improved Secchi depth in REANA suggests that the simulation of organic nutrient fluxes between the deep and surface layers is now more realistic. This is an

additional prerequisite for the calculation of nutrient budgets that will be presented in the following sub-sections.

## 5.5 Mean horizontal circulation of nutrients

Nutrient transport directly affects the biogeochemical cycles and the eutrophication of the Baltic Sea. Based on the reanalyzed simulation, the annual vertically averaged net DIN and DIP transports, as well as DIP persistency

are presented in Fig. 7. The persistency of the net transports is defined, for instance, by Eilola et al. (2012). One should note that the results by Eilola et al. (2012) are based upon 30-year averages for the control period 1978-2007 of a down-scaled climate scenario from a global circulation model. Similar calculations of transports and sources and sinks will therefore be briefly presented in the present study, since the present results better represent the hindcast period when the model is forced by the assimilated atmospheric (ERA–40) and Baltic Sea data

(REANA). DIP has the largest transports in the central parts of the Baltic proper, with high persistency because its concentrations are generally greater in deeper rather than in shallower areas. In the Bornholm Basin and the eastern parts of the central Baltic proper, cyclonic circulation patterns are found. In the western parts of the central Baltic proper, southward transports prevail. Relatively large amplitudes of transports of DIP are also found in the north-western Gotland Basin, in the southern Bornholm Basin, and through the Slupsk Channel

connecting the Bornholm Basin and Gotland Basin. Similar transport patterns are also found for DIN, OrgP and





OrgN. In contrast to Eilola et al. (2012), DIN, DIP, OrgP transports and their persistency are obviously stronger, although the overall patterns are similar. For example, in Eilola et al. (2012, their Fig. 1), large DIN transports appear in the southern Baltic proper and the Bornholm Basin. Similar differences are also found in both DIP and OrgP transports.

## 5.6 Internal nutrient sources and sinks

The horizontal distribution of areas with sources and sinks of phosphorus and nitrogen are illustrated in Fig. 8. A net inflow of nutrients to an area (import) is defined as a sink and counted as positive, while net outflow is defined as a source (export) and counted as negative (Eilola et al., 2012). Source areas of DIP generally coincide with sink areas of OrgP, and vice versa. This is also partly true for DIN and OrgN, but the sink for DIN has a large contribution from denitrification that transfers DIN to dissolved $N_2$. The difference between phosphorus and nitrogen sources and sinks is oxygen dependent, because the removal of N is enhanced at lower oxygen concentrations, while the sediment phosphorus sink is weakened (e.g., Savchuk, 2010). Sediments may even temporarily become a source under anoxic conditions, when older mineral-bound P can be released to the overlying water. Source areas of DIN are mainly found in the deeper parts of the Arkona Basin and Bornholm Basin. In contradiction to Eilola et al. (2012), areas with DIN export are also found at the southern and eastern coasts as well as at some small local regions in the inner parts of the Baltic proper. The largest DIP source occurs in the eastern parts of the Gotland Basin as well as in the deepest parts of the Bornholm Basin and Arkona Basin, whereas the largest sink of OrgP occurs in the central Baltic proper.

The main sources of DIP are generally found in regions where water depth is greater than 70 m (in other words below the permanent halocline in the Baltic proper), while the main sources of OrgP (and OrgN) are found in areas shallower than about 30–40 m (see also Fig. 9). Indeed, DIP export is largest in areas with a water depth between 70 and 100 m, and decreases towards greater water depths (Fig. 9). The magnitude of DIP imports and exports are greater than in Eilola at al. (2012), and there is pronounced import of DIP in the western part of the Eastern Gotland Basin below 100 m (Fig. 8) that is not as significant in Eilola et al. (2012). This, and the larger variability of DIN imports and exports, indicates that there is a higher degree of small-scale localized transport and production patterns that are not captured by Eilola et al. (2012). Main sinks of DIN are found in the deeper areas, but significant sinks are also seen in shallow areas and water depths of about 60m.



According to the accumulated import (Fig. 9), the magnitude of the DIP export is greater than that of the DIP import. This indicates that not all of the supply of phosphorus from land and atmosphere is retained within the
Baltic proper. For DIN, however, we may notice a very small net export from the Baltic proper to adjacent subbasins, while for OrgP and OrgN, imports and exports are almost balanced (Fig. 9). The nitrogen and phosphorus supply from land is implemented in sea areas with a bottom depth usually of 6 m. This is where the river mouths are located in the model.

There is a large import of DIP to areas with a depth range between 40–70 m (Fig. 9). This import does not
show a counter-part in the export of OrgP in Fig. 9.  This result might be explained by local processes causing the uptake and deposition of DIP. There is an import of DIN to these areas that together with nitrogen fixation and sediment–water fluxes of DIN may support local production of organic matter. As the assimilation of salinity observations result in a deeper halocline (Liu et al., 2014), the bottom water at a depth range of 40–70 m contains higher oxygen concentrations than in the simulation without data assimilation. Hence, in the REANA simulation
of this study, more phosphorus is taken up by the sediments at 40–70 m than in the simulation by Eilola et al. (2012). The phosphorus sink may also be partly caused by oxygen dependent water–sediment fluxes that bind DIP to ironbound phosphorus in oxic sediments (Almroth et al., 2015). This effect is not simulated, but may be accounted for by the transports adjusted by DIP concentrations assimilated to real observations in REANA. The relative importance of different processes is, however, not possible to evaluate from the reanalysis data set.

A partly opposite vertical exchange profile is found for OrgP. Coastal areas with a water depth of up to 40 m are exporting organic phosphorus, whereas deeper areas import OrgP. Production in the coastal zone of the Baltic proper and sedimentation in the open sea is almost balanced.

The largest export of DIN occurs due to rivers in the very shallow coastal zone. The magnitude of DIN imports and exports in areas with greater water depths are much smaller. Obviously, DIN supplied from land is already
consumed in the coastal zone (Voss et al., 2005; Almroth-Rosell et al., 2011) and, consequently, only a minor fraction of the nitrogen supplied to the shallow area can continuously reach regions deeper than 100 m (Eilola et al., 2012; Radtke et al., 2012). The present results show, however, an export contribution from DIN sources in deeper areas (e.g. 60–90 m depths) that may have been caused by reduced denitrification efficiency of oxidized sediments in the REANA simulation compared to Eilola et al. (2012).

**5.7 Nutrient budgets of subbasins**



The Baltic Sea is divided into seven subbasins according to the selected sections, which form the borders of the subbasins (Fig. 1). We calculate nutrient budgets for each of the subbasins from the reanalysis taking the nutrient supply from land and from the atmosphere into account (Fig. 10 and 11). The largest annual external phosphorus load occurs in the Baltic proper and amounts to 23.2 kton yr$^{-1}$ (Fig. 10). In addition, in the Baltic proper the

largest annual phosphorus sink of 12.7 kton yr$^{-1}$ is also found. The tendencies of phosphorus in the various subbasins differ. Whereas during the period from 1970–1999 the phosphorus content in the Gulf of Finland and Bothnian Bay increased, we found decreasing content in the Gulf of Riga, Baltic proper and Danish Straits (Table 1). In the Bothnian Sea, the difference between external supply and internal sink of phosphorus is equal to the net transport into the Bothnian Sea. The large burial of phosphorus in the Bothnian Sea is noteworthy. Largest export

and import of phosphorus between subbasins are found for the exchange between the Baltic proper and the Gulf of Finland, which amount to 54.7 and 50.7 kton yr$^{-1}$, respectively. However, the largest net exchange appears between the Baltic proper and Bothnian Sea. It is also found that the Baltic proper exports more phosphorus to each of its neighboring subbasins than it imports, except for the Gulf of Riga. The annual net phosphorus exported from the Baltic proper into the Danish Straits, the Bothnian Sea, the Gulf of Finland and Gulf of Riga

amounts to 2.4, 7.1, 4.0, and -1.4 kton yr$^{-1}$, respectively. The exchange of phosphorus between the Baltic proper and the Gulf of Riga is smallest relative to the other three neighboring subbasins. Further, we found that the net transport, import and export of phosphorus into the Bothnian Bay are smallest relative to the other subbasins.

Nitrogen transports between Baltic Sea subbasins are different compared to phosphorus transports (Fig. 11). For example, the Baltic proper has greater nitrogen sinks than external sources, while the Gulf of Riga has the same

external supply of nitrogen as the internal sink of nitrogen. Further, the nitrogen content in the Bothnian Sea decreased during the period from 1970–1999. We also found relatively large net transports of nitrogen from the Gulf of Riga into the Baltic proper. This is mainly explained by the relatively high nitrate concentrations in the Gulf of Riga relative to other subbasins.

To further analyze the variability of the budget of the reanalyzed nutrients, Fig. 12 provides the cross sectional,

integrated nutrient flows in the different subbasins. Here the eastward and northward net transports are, by definition, positive. Obviously, horizontally integrated nutrient flows vary significantly in space according to the nutrient loads from land. The inflows and outflows also vary depending on the depth of the water column and nutrient concentrations that influence the vertically integrated mass fluxes. In general, the magnitude of nutrient transports declines along transect A from south to north. For instance, the largest annual inflow of nitrogen in the





Baltic proper reaches 784 kton yr$^{-1}$, while it is only 266 and 173 kton yr$^{-1}$ for the Bothnian Sea and Bothnian Bay, respectively. The net flow and outflow of both nitrogen and phosphorus are similar in their spatial variations.

In the Baltic proper, inflows and outflow as well as the net northward flow of phosphorus increase from the south until a section along 56.8° N; they then remain about constant until a section along 58.7° N, and thereafter decrease rapidly further to the north. This indicates that major sources are located in the south where the large

rivers pour their loads into the Baltic Sea, while the major net sinks are mainly found in the northern parts of the Baltic proper. The behavior of net northward flow of nitrogen is different. Nitrogen transports decrease constantly with increasing latitude because the major sink (i.e. denitrification) works differently for nitrogen than for phosphorus, which is retained mainly by burial in the sediments. The net northward flow decreases at the latitude of the Gulf of Finland where phosphorus (and nitrogen) is transported towards the Gulf, as seen in transect C.

In the Arkona and Bornholm basins, nitrogen and phosphorus transports increase from the west to the east. Due to the nitrogen load from the Oder River, the inflow of nitrogen increases significantly at the border between the Arkona and Bornholm basins, whereas the outflow does not show any discontinuity. As a result, the net flow of nitrogen shows an accelerated increase. The situation for phosphorus in the Arkona and Bornholm basins is different compared to the nitrogen transports because in- and outflow, as well as the net flow, change direction.

The phosphorus loads from the Oder River turn the outflow in the western parts into an inflow of phosphorus in the eastern parts.

In the Gulf of Finland, in– and outflows generally decline from the west to east. In the entrance of the Gulf of Finland, the inflows of nutrients are almost zero. The largest net flows of nutrients (outflow) appear at the entrance of the Gulf of Finland, with a magnitude of 65 kton yr$^{-1}$ for nitrogen and 5.1 kton yr$^{-1}$ for phosphorus,

respectively. Although the net flows of phosphorus and nitrogen change their directions in the Gulf of Finland, the location of the transition differs. For nitrogen, this change in direction is closer to the entrance than for phosphorus. These results indicate that the large supply of nutrients from the Neva River are accumulated or removed within the Gulf of Finland.

6 **Discussion**

380 The initial conditions in the FREE run used in this study as a reference are imperfect, causing large biases compared to observations. The reason is that the nutrient pools in the sediments have not been spun up appropriately. As a consequence, phosphate concentrations in FREE are higher than observed concentrations at





all depths (Fig. 5). The biases in surface phosphate concentrations between model results and observations can influence the seasonal primary production. In REANA, however, from the beginning of the experiment, the biases are already significantly reduced and remain relatively small during the integration compared to FREE. The biases of phosphate reduce with time both in the FREE and REANA runs. Hence, this indicates a need of new initial conditions of the sediments and perhaps a recalibration of the biogeochemical model, as was also discussed by Liu et al. (2014).

Fu (2013) estimated the volume and salt transports during the 2003 MBI with 3DVAR in the Baltic Sea. In the present study, we estimate the impact of the data assimilation based on the EnOI method on the net volume and nutrient transports as well as calculate budgets for major subbasins of the Baltic Sea. The volume transports obtained with different assimilation methods may be different. The sea level in Fu (2013) is kept constant in the assimilation process, while sea level in this study is varying accordingly during the assimilation of temperature and salinity based upon the statistical covariances. The variability of sea level may enhance the barotropic flow, which is one of the reasons for the differences in net volume transport in the two simulations. However, transports within the subbasin are also indirectly affected by the interaction of baroclinicity and topography.

The in- and outflows of phosphorus simulated in REANA are greater than the results by Wulff and Stigebrandt (1989), Savchuk (2005) and Savchuk and Wulff (2007). However, the net transports of phosphorus are similar between our results and these earlier studies. Moreover, the nitrogen budgets are somewhat lower than the results of earlier studies, especially in the Baltic proper. However, it should be kept in mind that the above mentioned studies estimated the nutrient budgets from mass balance models together with inter-basin transport calculations based upon Knudsen's formulae to calculate nutrient budgets of the Baltic Sea (see, e.g. Savchuk, 2005). Hence, with this approach the wind driven circulation is not included. Consequently, our results show greater in- and outflows between subbasins and consider nutrient transports caused by Ekman dynamics. Obviously, there are limitations in calculations of previous studies. Despite overall uncertainties that also limit the reliability of our results, like incomplete understanding of selected biogeochemical processes (e.g. nitrogen fixation), lacking information of sediment parameters, and under-sampled observations in space and time, our approach has the advantage of using both high-resolution modeling and all available observations made over a 30-year period. Our model results consider the complete set of primitive equations in high-resolution, taking into account not only the volume and salt conservation of subbasins according to Knudsen's formulae, but also the wind-driven circulation between and within subbasins. Hence, we have, for the first time, the potential to quantify spatial transport





patterns with high confidence even within subbasins, as in the exchange of nutrients between the coastal zone and the open sea.

Further, our study covers a different time period compared to the studies by Wulff and Stigebrandt (1989),
Savchuk (2005) and Savchuk and Wulff (2007). This is important because nutrient concentrations and related budgets vary in time and space. For example, during the period 1970–1999, HELCOM (2013) showed that the total phosphorus (TP) concentration has generally decreased in the Bothnian Bay and has increased in the Gulf of Riga. However, these changes in TP concentrations were not monotonous. For example, the TP concentration has obviously increased during the period 1970–1976 in the Bothnian Bay. While, in the Bothnian Sea, TP
concentration has increased during the period 1970–1983 and decreased during the period 1990–1999. Similarly, changes in total nitrogen (TN) concentration differed during different periods. Nutrient budgets (Figs. 10 and 11) of subbasins are time-averaged and represent in our study the overall results of the period 1970−1999. Hence, it is not surprising that other studies got very different results. For example, Savchuk and Wulff (2007) found lower TP concentration in the Bothnian Bay relative to our results because their experiment covered the period 1997−
2003 and the DIP concentration has generally decreased after the year 1985 (HELCOM, 2013).

Eutrophication of the Baltic Sea is directly affected by the long-term evolution of external nutrient supply. The external nutrient input has three components (waterborne land loads, direct point sources at the coasts, and atmospheric depositions) which are associated with the biogeochemical dynamics of the Baltic Sea. Savchuk and Wulff (2007) used the historical loads of total phosphorus based on filtered samples. However, their total
phosphorus load, for example to the Gulf of Finland, is underestimated because the particulate phosphorus fraction is neglected (Savchuk et al., 2012). In our study, we used the reconstructed external nutrient input data by Savchuk et al. (2012). Therefore, the phosphorus supply into the Gulf of Finland is greater in our study compared to Savchuk and Wulff (2007). The greater phosphorus supply changes the phosphorus content and phosphorus concentration in the Gulf of Finland. This is another reason why phosphorus transports between the Gulf of
Finland and the Baltic proper in our study are greater than the transports calculated by Savchuk (2005) and Savchuk and Wulff (2007).

Gustafsson et al. (2012) used a process-oriented model that resolves the Baltic Sea spatially in 13 dynamically interconnected and horizontally integrated subbasins with high vertical resolution to reconstruct the temporal evolution of eutrophication for 1850–2006. Savchuk (2005) and Savchuk and Wulff (2007) applied mass balance
models as mentioned above to calculate nutrient budgets of the Baltic Sea. The results of all these models depend on the locations of the subbasin borders that are for some subbasins only arbitrarily defined. Using a high-



resolution circulation model, we showed that nutrient flows within the subbasins vary considerably (Fig. 12). For instance, we found east- and westward net transports of nitrogen between the Baltic proper and Gulf of Finland depending on border locations at 23.2° and 24.0° E, respectively.

**7 Summary and Conclusion**

For the first time, a multi-decadal, high-resolution reanalysis of physical (temperature and salinity) and biogeochemical variables (oxygen, nitrate, phosphate and ammonium) for the Baltic Sea was presented. The reanalysis covers the period from 1970–1999. A "weakly coupled" assimilation scheme using the EnOI method was used to assimilate all available physical and biogeochemical observations into a high-resolution circulation
model of the Baltic Sea.

Both assimilated and independent observations collected from different databases are used to evaluate the reanalysis results (REANA). Based on the model–data comparison presented in this study, we found that the model results without data assimilation (FREE) exhibit significant biases in both oxygen and nutrients. The reasons for these biases are not totally understood yet, although it is speculated that the main reasons might be
related to limitations of model parameterizations and the imperfect initial conditions. Based on the calculation of the overall RMSD of oxygen and nutrient concentrations between model results and not-yet-assimilated observations, the results in REANA are considerably better than those in FREE. The total RMSD of the oxygen, nitrate, phosphate and ammonium is reduced respectively by 0.84 mL L$^{-1}$, 0.99 mmol m$^{-3}$, 0.88 mmol m$^{-3}$, 0.52 mmol m$^{-3}$. This means that the overall qualities of simulated oxygen, nitrate, phosphate, and ammonium
concentration are improved by 59, 46, 78 and 45%, respectively. These results demonstrate the power of the applied assimilation scheme.

The observation information entering the model affects the oxygen dependent dynamics of biogeochemical cycles significantly. As examples, we presented results of mean seasonal cycles of nutrients, the spatial surface distributions of DIN, DIP and DIN:DIP of the entire Baltic Sea, and the interannual variations in Secchi depth in
various subbasins. According to Liu et al. (2014), the improved biogeochemical cycles are due to both improved simulation of physical (e.g. vertical stratification) and biogeochemical parameters (e.g. sources and sinks).

Based on the reanalysis simulation, we analyzed nutrient transports in the Baltic Sea. We found that vertically integrated nutrient transports, to a large extent, follow the general horizontal water circulation, and vary spatially to a large extent. In particular, large nutrient transports were found in the Eastern Gotland Basin, in the Bornholm



Basin, in the Slupsk Channel and in the north-western Gotland Basin. The persistencies of nutrient transports are greater in the eastern and southern than in the northern and western Baltic Sea.

The horizontal distributions of sources and sinks of inorganic and organic nutrients show large spatial variations and may be partly explained by (1) the external supply of nutrients from land, (2) the topography controlling horizontal nutrient exchange between subbasins and between the coastal zone and the open sea, and

(3) vertical stratification that determines redox conditions at the sea floor. The latter is important for the water–sediment fluxes of nutrients, and consequently for burial of nutrients in the sediments. The reanalysis results suggest that in the Baltic proper, in most areas with a water depth less than the depth of the permanent halocline at about 70–80 m, DIP is imported and transformed either to OrgP, or buried in the sediments in water depths greater than the wave-induced zone at 40–70 m. Whether the latter is an artefact of the assimilation method or a

real sink is unclear. On the other hand, in areas with greater water depth, DIP is exported (e.g. released from the sediments under anoxic conditions). Overall, the Baltic proper exports DIP to neighboring subbasins.

The nitrogen cycle is very different compared to the phosphorus cycle. The shallow coastal zone with water depths less than 10 m plays an outstanding role for DIN, because within it, large exports occur due to supplies from land. Most of the exported DIN is removed in shallow waters, mainly by denitrification, while at greater

depths imports and exports of DIN are much smaller, indicating the important role of the coastal zone for nitrogen removal.

Detailed nitrogen and phosphorus budgets suggest that nutrient cycles in the various subbasins are controlled by different processes and show different response to external loads and internal sources and sinks. In particular, the Baltic proper is the subbasin with the largest nutrient exchanges with its surrounding subbasins. The Baltic

proper exports phosphorus to all subbasins except the Gulf of Riga. Similarly, the Baltic proper also exports nitrogen to all subbasins except to the Gulf of Riga and Danish Straits. In this subbasin, the largest internal sink of all subbasins was also found. Noteworthy is the relatively large net export of phosphorus from the Baltic proper into the Bothnian Sea, where the respective second largest sinks for both phosphorus and nitrogen were found. This finding is in agreement with previous studies. For the budgets of the subbasins, it is important where

the borders of the subbasins are located, because net transports may change sign with the location of the border. For instance, in the entrance of the Gulf of Finland, the net phosphorus transport from the Baltic proper is directed eastward, but changes direction at about 26ºE. Further to the east, the net phosphorus transport is directed westward.



**Acknowledgements**

The research presented in this study is part of the Baltic Earth programme (Earth System Science for the Baltic Sea region, see http://www.baltic.earth), and was funded by the Swedish Research Council for Environment, Agricultural Sciences and Spatial Planning (FORMAS) within the projects "Impact of accelerated future global mean sea level rise on the phosphorus cycle in the Baltic Sea" (grant no. 214-2009-577),"Impact of changing climate on circulation and biogeochemical cycles of the integrated North Sea and Baltic Sea system" (grant no.

214-2010-1575), and "Cyanobacteria life cycles and nitrogen fixation in historical reconstructions and future climate scenarios (1850-2100) of the Baltic Sea" (grant no. 214-2013-1449), as well as by the Swedish Research Council within the project "Reconstructing and projecting Baltic Sea climate variability 1850-2100" (grant no. 2012-2017).

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



Table 1. The 30-year mean tendencies of total phosphorus and nitrogen in Baltic subbasins. Names of the
670   subbasins are the Kattegat (KT), Danish Straits (DS), the Baltic proper (BP), the Gulf of Riga (GR), the Gulf of
Finland (GF), the Bothnian Sea (BS), and the Bothnian Bay (BB).

| $10^3$ton yr$^{-1}$ | KT | DS | BP | GR | GF | BS | BB |
|---|---|---|---|---|---|---|---|
| $\Delta$P | 4.3 | -3.8 | -1.6 | -1.4 | 5.9 | 0 | 0.8 |
| $\Delta$N | -3 | -46 | -133 | -12 | 21 | -29 | -5 |




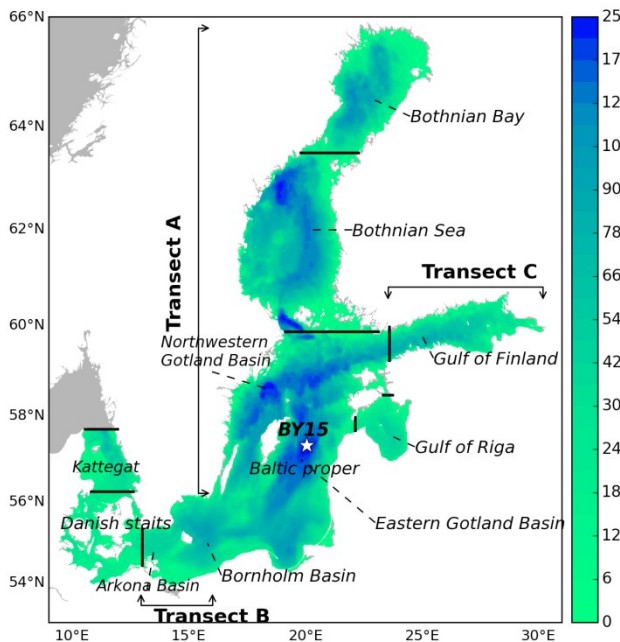

Figure 1. The bathymetry of the model (depth in m). The border locations of subbasins of the Baltic Sea used in this study are shown by the black lines, and the BY15 station is shown by the white star.





675

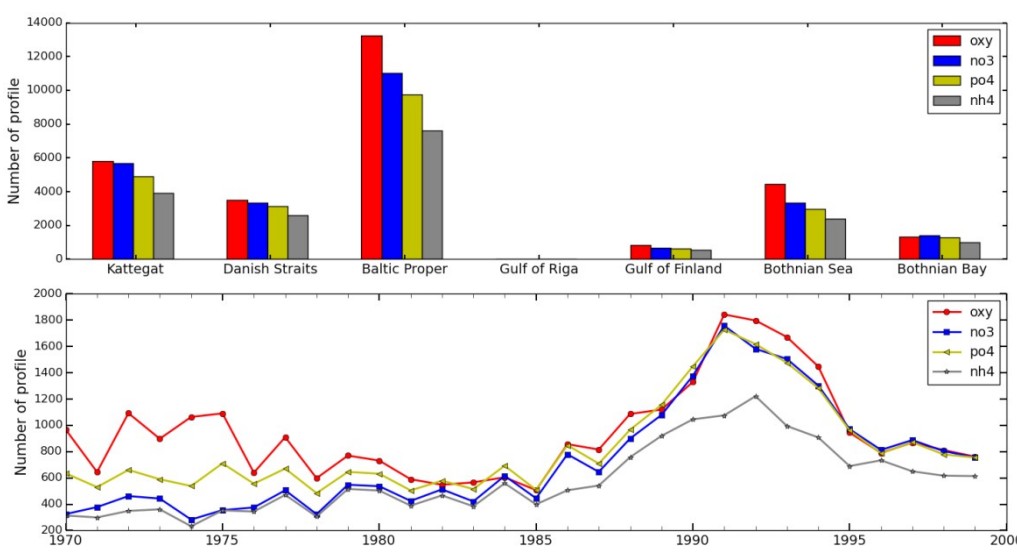

Figure 2. Number of observations in different subbasins (upper panel) and annual number of observations from 1970-1999 (bottom panel).



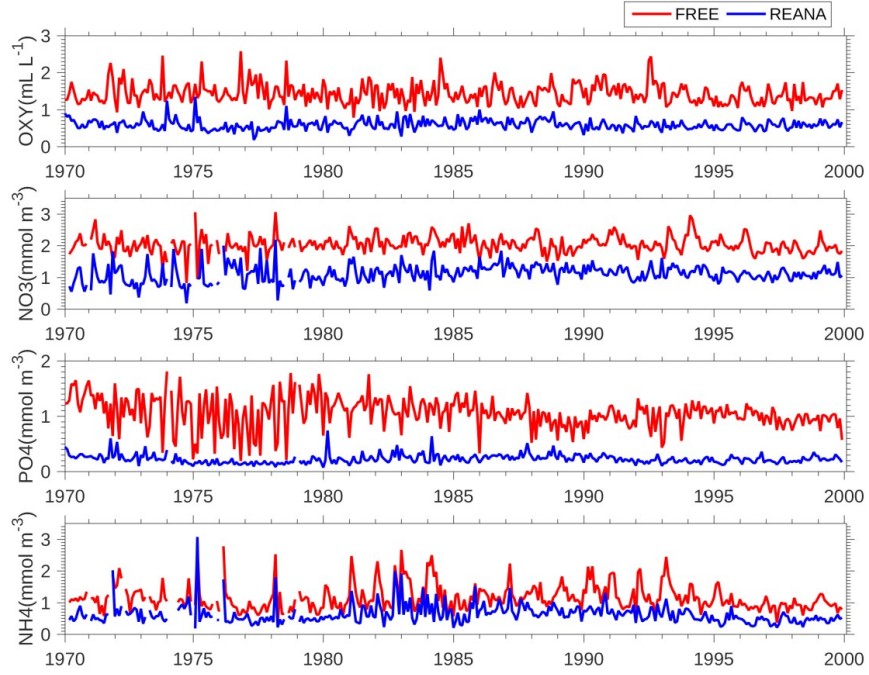

680  Figure 3. Monthly mean root mean square deviation (RMSD) between model results and observations for oxygen, nitrate, phosphate and ammonium in FREE (red) and REANA (blue).





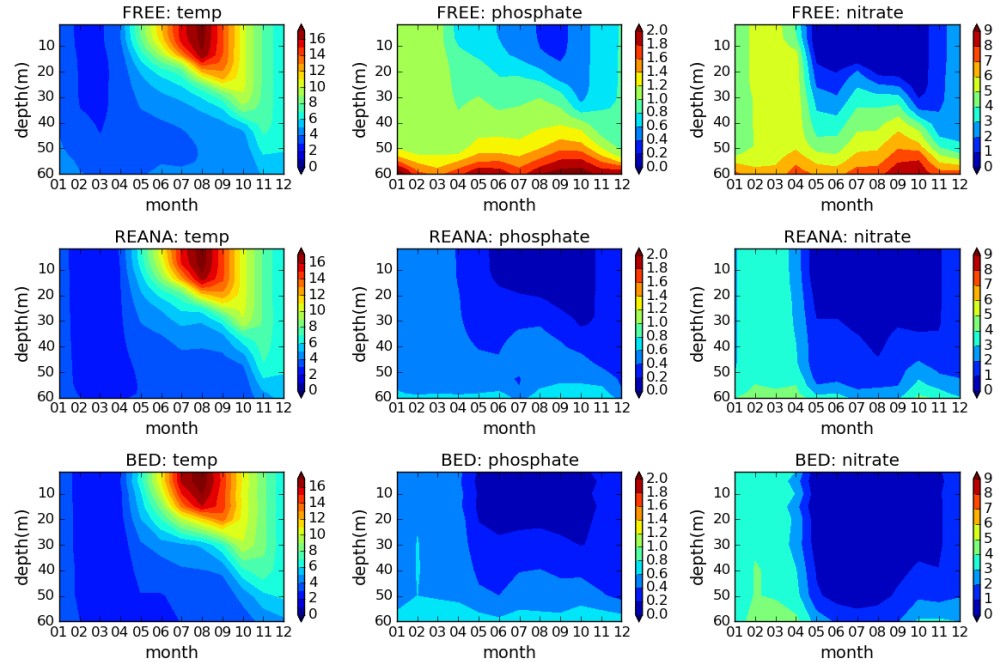

Figure 4. The annual cycle of monthly average (1970–1999) temperature (°C), phosphate concentration (mmol m$^{-3}$), and nitrate concentration (mmol m$^{-3}$) at BY15 for FREE (row 1), REANA (row 2), and BED data (row 3), respectively.




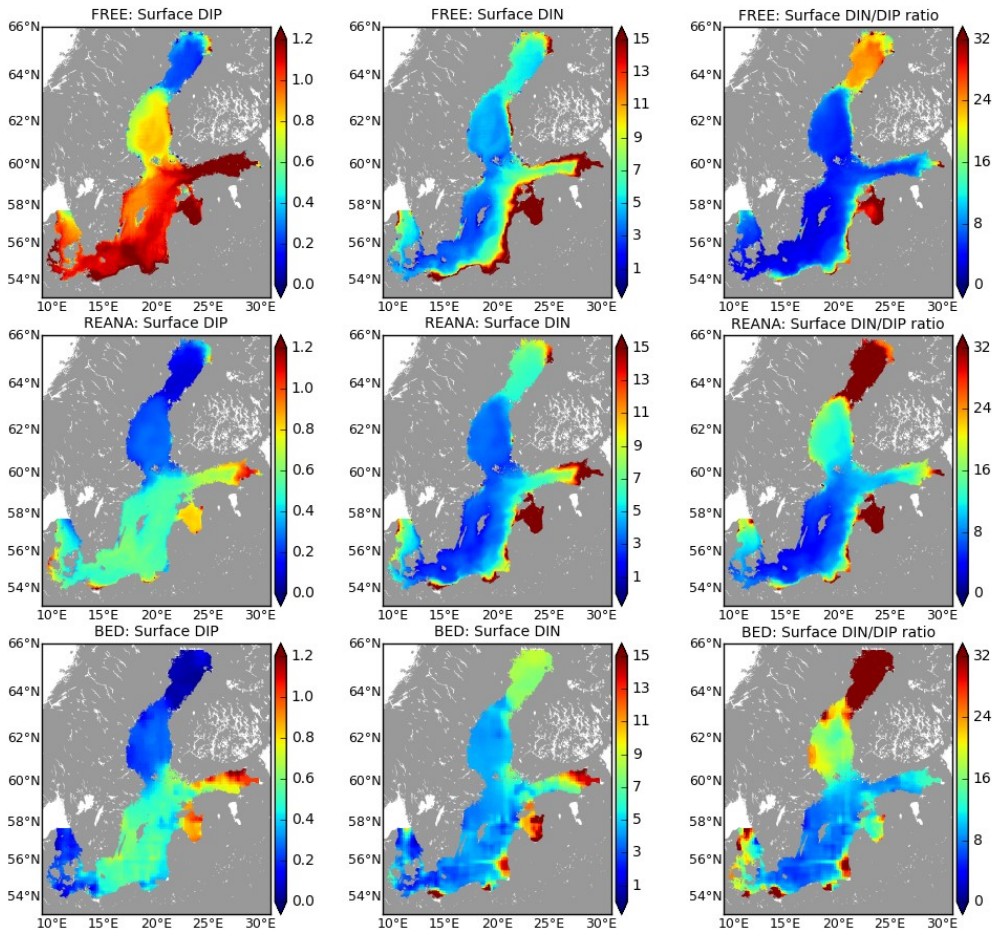

Figure 5. Monthly (March) mean (1970–1999) surface layer (0–10 m) concentrations of DIP (mmol m$^{-3}$) (left), DIN (mmol m$^{-3}$) (middle), and the corresponding DIN to DIP ratio (right). Results from FREE, REANA and BED are shown from above in rows 1, 2 and 3, respectively.




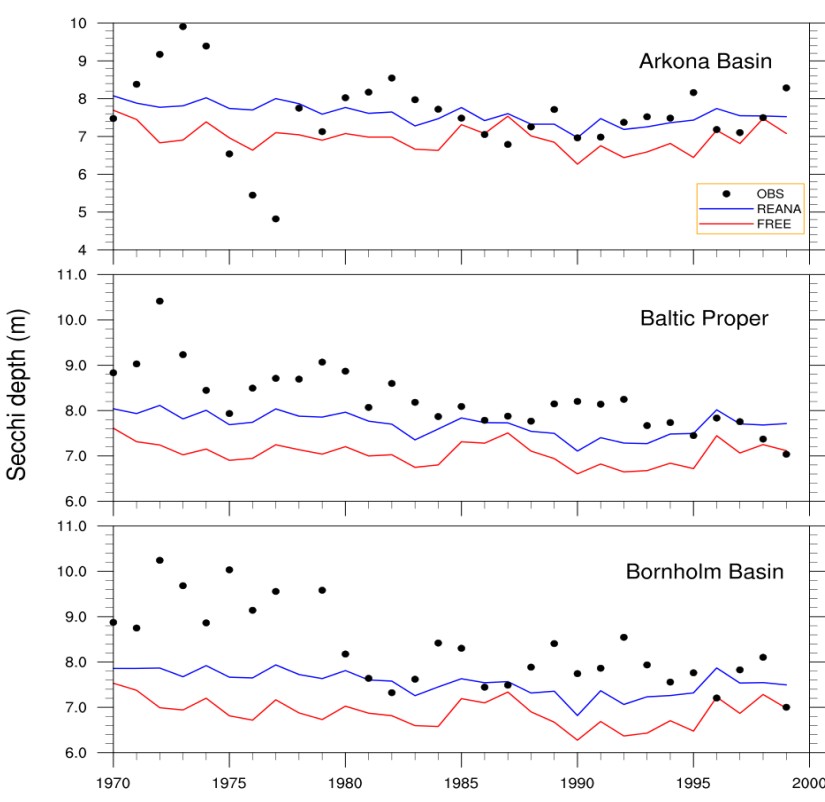

690

Figure 6. Time series of the annual average Secchi depths for the observations (black) and the simulation of REANA (blue) and FREE (red) in the Baltic Sea, respectively, for the period 1970-1999.





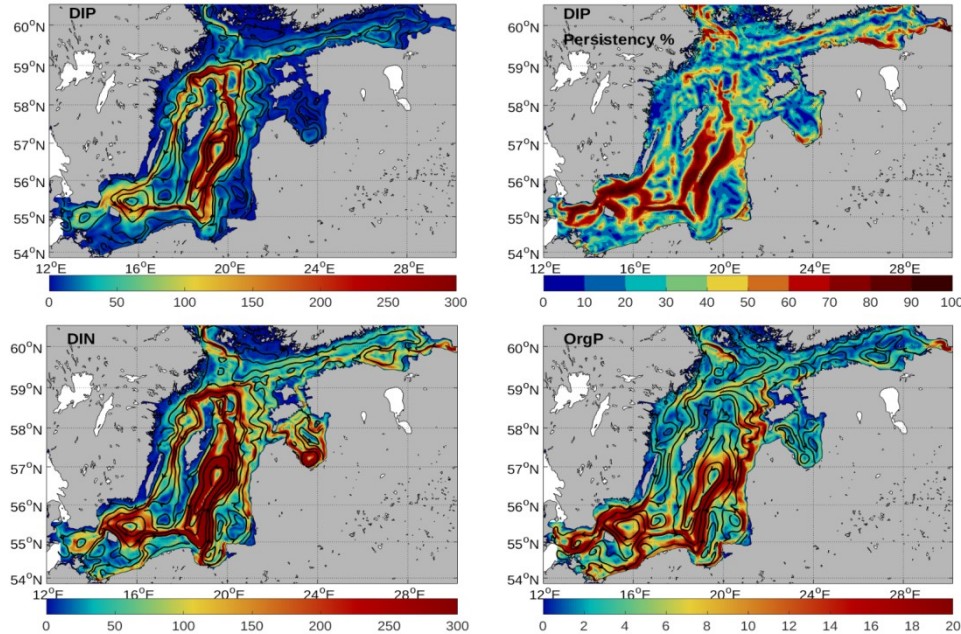

Figure 7. Annual average net DIN (a), DIP (b) and OrgP (c) transports and the corresponding DIP persistency (d) shown for REANA. The black solid lines with arrows show the streamlines and direction of transports. The magnitude of transports (ton km$^{-1}$ month$^{-1}$) and the persistency (%) are shown by the background color. The corresponding values are shown in the colored bars.




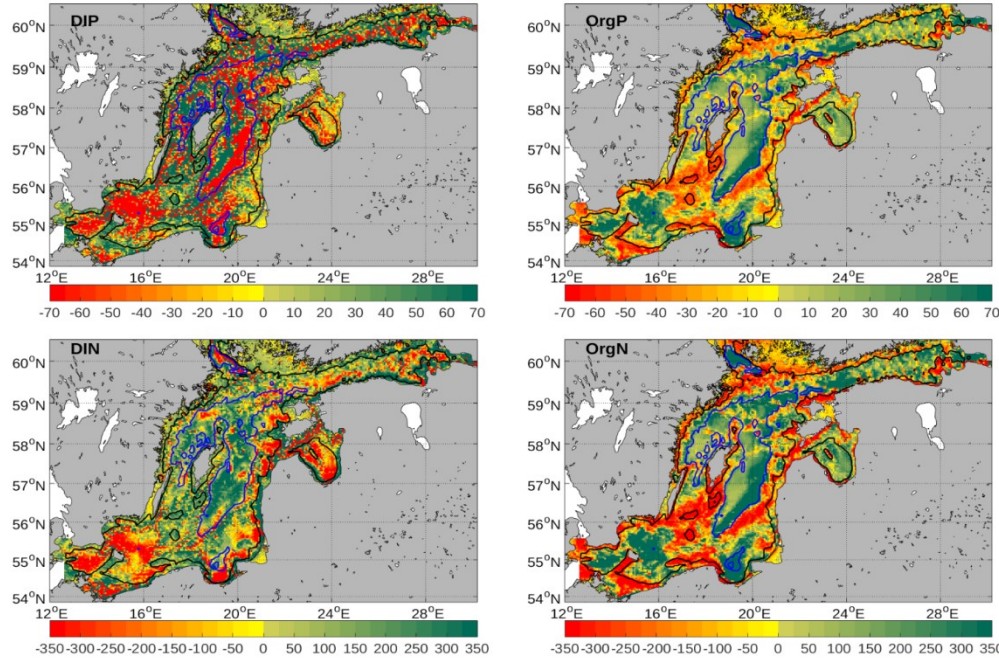

Figure 8. Spatial distributions of annual average import of DIN, OrgN, DIP and OrgP, respectively. The magnitude of import (ton km$^{-2}$ month$^{-1}$) is shown by the background color with the corresponding values shown in the right color bar. Green color denotes positive values (import), and yellow to red colors denote negative values (export). The black and blue lines show 30 and 100 m depth contours of the model.




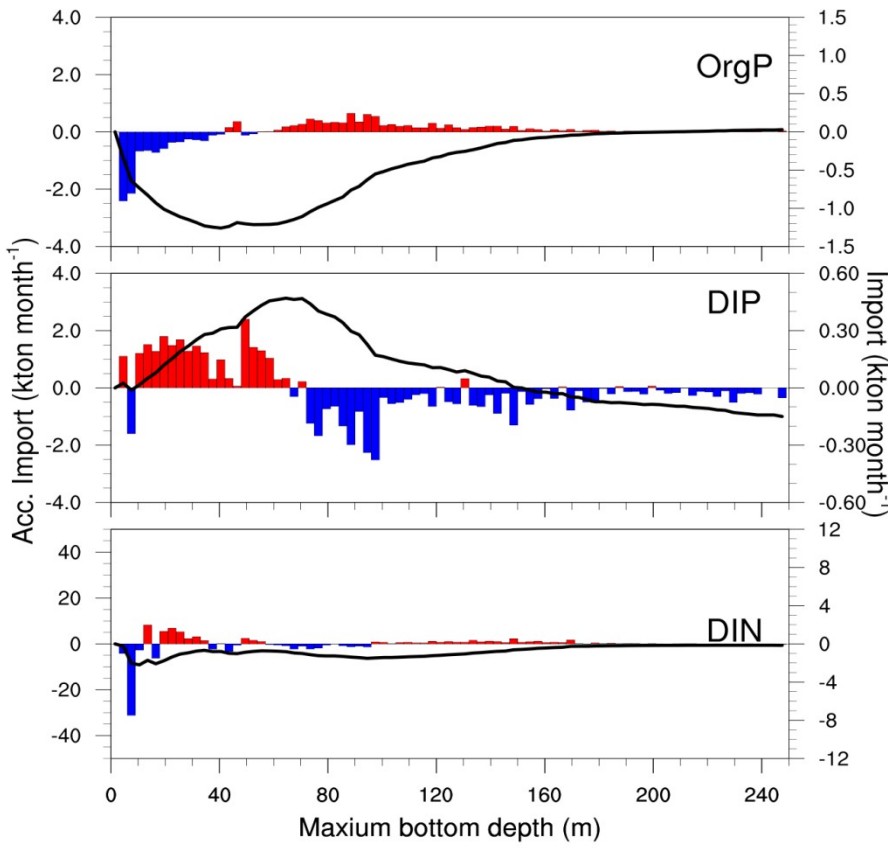

Figure 9. Annual average accumulated imports (black lines) and imports of OrgP, DIP, and DIN (color bars) to
regions with the same depth in the Baltic proper.



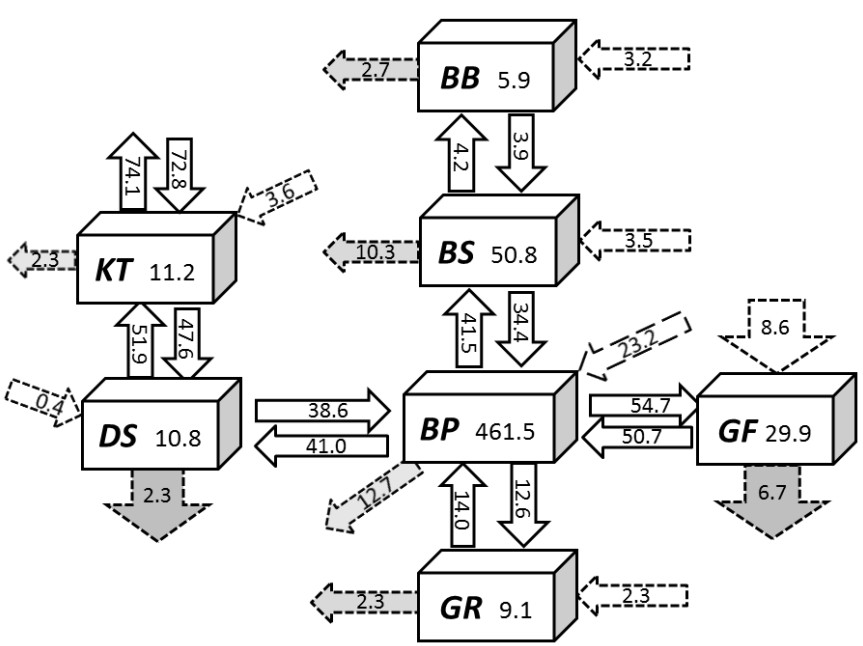

Figure 10. The annual total phosphorus budgets of the Baltic Sea averaged for the period 1970–1999. The average total amounts are in $10^3$ tons, and transport flows and sink/source fluxes are in $10^3$ tons year$^{-1}$. External nutrient inputs are separated into terrestrial and atmospheric sources. Terrestrial loads are reduced by phosphorus retentions for the coastal zones.




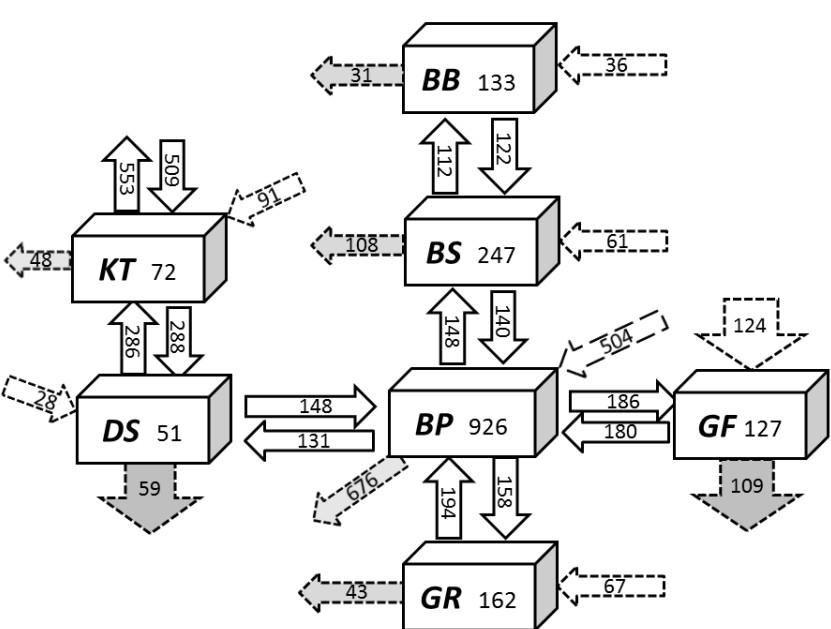

Figure 11. The same as Figure 10, but for nitrogen.

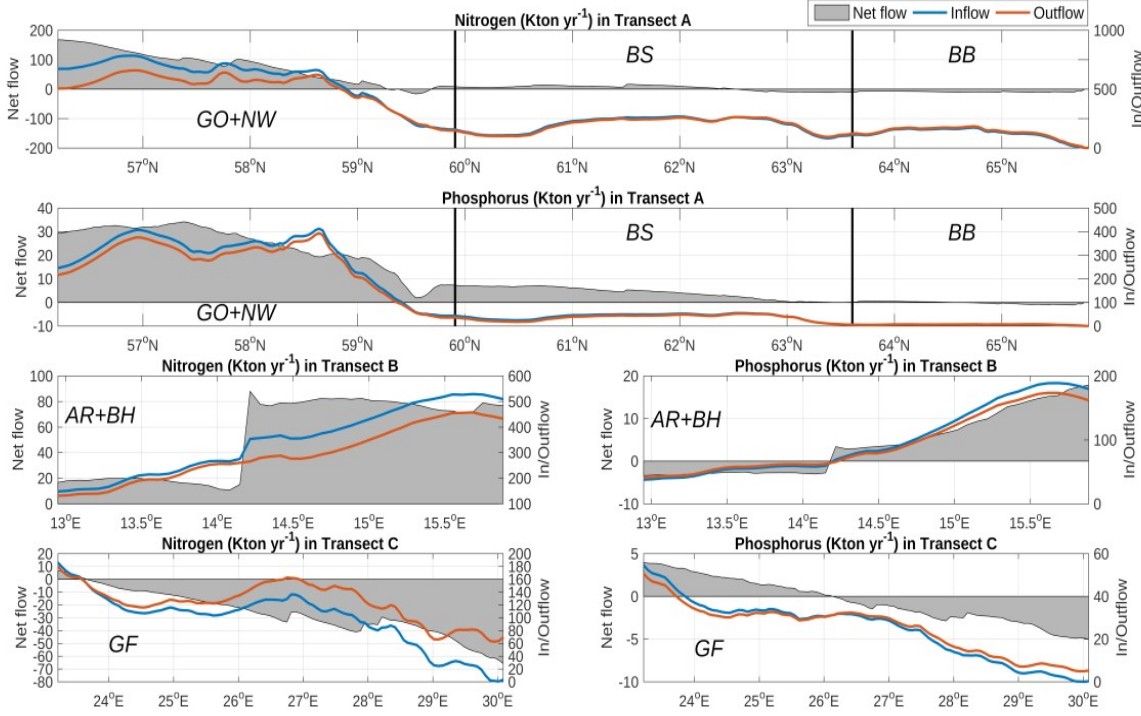

Figure 12. The annual average fluxes of nitrogen (in $10^3$ ton N yr$^{-1}$) and phosphorus (in $10^3$ ton P yr$^{-1}$) as a function of the cross sections along transects following the latitude and longitude in the Baltic subbasins. Northward and eastward fluxes are, by definition, positive. Here, AR, BH, GO, NW, GF, BS, and BB represent the Arkona Sea, Bornholm Sea, Eastern Gotland Basin, Northwestern Gotland Basin, Bothnian Sea and Bothnian Bay, respectively. Transect A summarizes fluxes from the southern Baltic proper to the Bothnian Bay. Transect B describes the Baltic Sea entrance area from the Arkona Basin to the Bornholm Basin, and transect C summarizes fluxes in the Gulf of Finland.