# Peer review of "Nutrient transports in the Baltic Sea - results from a 30-year physical-biogeochemical reanalysis"

_Biogeosciences, 2016_

## Referee Comment (RC1) · O.P. Savchuk (Referee) · 1 Oct 2016

The study deals with application of data assimilation approach to reconstruction of long-term dynamics of 3D nutrient fields as a base for analysis of nutrient transport processes in the Baltic Sea. Both the approach and obtained results are significantly novel in methodological and geographical senses to deserve publishing in "Biogeosciences". However, scientific and presentation qualities should be substantially improved by the major revision of the manuscript along the lines suggested below.

1. General comments and suggestions

1.1 Objectives and applicability. The assimilation of whatever available data is fully

justified for an improvement of short-term forecasting of hydrophysical fields aiming at the search-and-rescue operations, propagation and expansion of catastrophic spills as well as management of the maritime activity. However, its applicability for long-term hindcasts of biogeochemical phenomena and properties requires careful consideration and clear explanation of the purposes/objectives of the assimilation (why and what for). Such considerations and explanations should already be given in the Introduction section, with particular attention to the limitations, especially non-conservativeness of the approach (what can and cannot be done).

1.2 Artificial non-conservation. Biogeochemical variables are non-conservative by definition, while the entire models of biogeochemical cycles are usually designed as conservative, i.e. explicitly accounting for all the external and internal sources and sinks of the matter. In such models (including the implemented RCO-SCOBI system), the dynamics of simulated nutrient fields is determined by continuous, mutually adjusted interaction of physical transport and biogeochemical transformation processes. If these 4D fields (x, y, z, t) are not absolutely identical to the corresponding fields reconstructed from observations, then an every act of "correction" of simulated towards reconstructed fields during assimilation procedure would create in the model fictitious 3D sinks and sources of the matter not generated by either transport or transformation processes. These fictitious fluxes of nutrients are then included into biogeochemical cycles, thus making the model erroneously non-conservative. Evidently, the studies of eutrophication and biological productivity in general are particularly vulnerable for these effects of data assimilation. As can be deduced, for instance, from Figs. 3-5, such effects are quite substantial.

On the other hand, with a certain confidence in simulated transport agents (water currents and mixing) supported, e.g. by the plausible dynamics of "conservative" salinity (e.g. as in Liu et al. 2013), the "corrected" fields of nutrients could be used for improving simulation of nutrient transport processes. Here, again, the discussion on how such improvement would affect simulation of transformation processes and, in turn, would

be affected by them could significantly augment the scientific value of the paper. Also, the questions arises – could not the same results regarding transport processes been achieved just with the "observed" nutrient fields used for assimilation, without running and "jerking/correcting" the biogeochemical model.

In any case, the artificial non-conservativeness should be explicitly acknowledged and explained, its effects evaluated, presented, and discussed, in addition to- and, perhaps, together with analysis of biases by means of RMSD. The estimates of non-conservation and its spatial and temporal dynamics must be computed from a difference between model fields before and after acts of assimilation, starting from the initial conditions. Then the knowledge of needed "correction" can also be used in pinpointing possible deficiencies in the biogeochemical parameterizations.

1.3 Plausibility of the RCO-SCOBI model. The RCO-SCOBI model has been extensively used for forecasts (aka projections) of possible changes in the Baltic Sea biogeochemistry under different scenarios of driving forces, practically by the same authors. Therefore, the scientific value of the paper could be significantly increased by the discussion and speculations on how the model's deficiencies in simulation of transport flows and transformation fluxes, which are revealed due to the data assimilation, for instance, in the form of RMSD, could affect the predictions. Good starting point could be a statement at line 387.

1.4. Description and explanation of Methods. All the methods implemented in the manuscript must be described in more detail and, considering an intended expansion of the paper's coverage from the "hydrophysical" audience over the "Bio-Geo-Chemical" one, in somewhat more popular style.

Assimilation procedure. In addition to references to (Liu et al. 2013, 2014), several details, especially those important for magnitude and distribution of 4D fictitious fluxes, must be repeated and explicitly explained in this paper as well. The explanations should include, for instance, such details as: a) verbal description of procedure

for reconstruction of "observed" fields used further in assimilation and in calculation of RMSD in FREE and REAN experiments, b) spatially and temporally varying uncertainties of such fields determined by the scarcity and sparsity of observations, c) frequency of the assimilation acts and its possible effects on the difference between model and observation used in calculation of RSMD (Liu et al., 2014), and whatever else would be necessary for further presentation and discussion of issues from Comment 1.2 above. Without such clarifications, three sentences at lines 170-173 look as isolated abracadabra and might seem almost useless.

Nutrient transports, trends, and budgets. The exact definitions of all the nutrient transports, trends, and budgets measures and characteristics together with algorithms of their calculation, including derived units, should be clearly presented already in Methods. This will clarify possible confusions with the usage and interpretation of the terms vs. phenomena, commented in details below, in Section 2.

2. Specific comments and suggestions.

2.1 "Cycling" in the title and similar statements to that effect elsewhere Accordingly to comments 1.1-2 above, the non-conservative model cannot be used for comprehensive studies of nutrient CYCLING. Hence, the title should be modified – consider, please, something like "Nutrient TRANSPORTS in the Baltic ...." instead. Correspondingly, the usage of "cycling" and similar statements and expressions about transformation processes should be carefully revaluated throughout the entire text, for instance, at lines 80, 189, 310, 306-307, 362-363, 466, and throughout the entire Section 5.6,

2.2 Calculation of RMSD. Line 194 – What is the meaning of "overall" and "monthly mean" in "the overall monthly mean RMSDs" and how they were calculated – for how many fields per month? covering the entire Baltic? cell by cell for interpolated "observational" fields or only for cells with the real observations?

2.3 Nutrient transports. Explain and clarify, please, involved terms and interpretations – What does "net" (which is usually used with the word "exchange" and represents a difference between inputs/imports and outputs/exports) mean at lines 17, 259-260, 277, 300, 338, 356, 360, and 492; – Why some characteristics related to single grid cells or a grid "column" are called "net", has it something to do with the difference between in- and out- transport flows or/and is it meant to account for local changes due to transformations, causing difference between inflows to the cell (column) and outflows from it? For instance, at line 694 – How exactly the vertical averages and vertical integrals (e.g. line 259) have been computed? Why ANNUAL average is expressed in ton/ km/MONTH (Fig. 7, lines 694-697)? Would vertical averages multiplied by the depth of grid point be equal to vertical integrals? What is the point presenting/contrasting/comparing (e.g. in Fig. 7) vertically averaged transport for the locations with, for instance, 200 and 20 m depths? – Definitions and explanations for calculations of nutrient sources and sinks from integral transports would be helpful in understanding and interpretation of Section 5.6. Some consideration and discussion on how much the sinks and sources could depend on which transformation processes and how much they would be determined by fictitious fluxes might be useful too. Also, check the consistency of term's usage both in the text and, especially in legend to Figs. 8 and 9 (annual average IMPORT (transports?); again ANNUAL is expressed on per MONTH basis.

2.4 Nutrient budgets. Explain, please, how the budgets were computed: – How nutrient in- and outflows (as product of velocity and concentration) been obtained from integrals of continuous computations for period 1970-1999 or from averaging of monthly or annual integrals? – How have annual sink/sources been calculated? Have the transformation processes (sediment-water exchanges, burial, nitrogen fixation, denitrification) been accounted for? – How trends in Table 1 been estimated? What does P sources in the KT, GF, and BB (sic!) as well as N source in GF mean? – How the total amounts (pools) of nutrients were calculated, by averaging of which fields, integrated with which frequency?

These explanations are necessary but not sufficient for understanding how 30-year average annual "tendencies" (trends? deviations?) agree with pools? Most illustrative

are P sources. In BB, 0.8 Kt P/yr *30 yrs=24 Kt P comparing to the pool of 5.9 Kt P; in GF, 5.9 Kt P/yr *30 yrs=177 Kt P comparing to the pool of 29.9 Kt P. Where has such hefty P excess gone, accumulated in the sediments? Evidently, the changes of nutrient pools in sediments must be included into consideration as well regardless of how plausible they are.

– Legend to Figs. 10 and 11 says: "External nutrient inputs are separated into terrestrial and atmospheric sources. Terrestrial loads are reduced by phosphorus retentions for the coastal zones." However, external inputs are presented with single numbers. Is it a sum of terrestrial and atmospheric loads, then the word is "combined"? What is the coastal P retention, how it was estimated and which values were prescribed? Was N inputs treated in a similar way?

Similar explanations and considerations, starting from algorithm of calculation should be given also to horizontally integrated flows at transects (Fig.12, lines 349-378) with special attention paid to explanation of the purpose of their analysis in a view of complex picture of water circulation and nutrient transports in Fig. 7.

Considerations about possible contributions of transformation vs. fictitious processes would be appropriate in Section 5.7 or in discussion of presented results as well.

2.5 Secchi depth (see also comment for lines 185-186 below). The water transparency seasonal variations and long-term trends depend on too many factors that either are not included in the model (e.g. CDOM and SPM distribution and variation) or are determined by complicated feedbacks from transformation processes (e.g. primary production and sedimentation of decomposing organic matter) to be used as unequivocal indicator of improved simulation of the nutrient fields. In result, the related analysis (lines 250-253) looks weak and unconvincing, for instance, the decrease of inorganic nutrients should cause the decreased primary production and how realistic is that? Or is it a correct effect by the wrong reason? Therefore, I would recommend deleting consideration of Secchi depth from the paper entirely. However, if the authors will chose to

retain these considerations then a few words about how Secchi depth is estimated in the model (what it does and does not account for) would be useful for readers.

2.6 Presentation of pelagic and sediments pools. As it appears from Comments 2.4 and lines 380-388 in Discussion, presentation of pelagic and sediment nutrient pools could help to untangle several issues in interpretation of results

3. Minor things, technical corrections and language cosmetics.

Lines: 3 – I guess, it is Eilola not Eolila; 11-12 – What is "improvement in . . . concentrations"? Consider, please, something like "improved simulation/reproduction/imitation of concentrations" or similar; 33-34 – Perhaps, not as much "living conditions" as redox dependent biogeochemical processes; here the reference to (Conley et al., 2009) or/and (Savchuk, 2010) would be appropriate in addition to- or instead of (Fu, 2013) 50-54 – poor choice of words: ". . .of BIOLOGICAL formulations (either empirical or mechanistic) to UPDATE biogeochemical concentrations" that sounds as (physical) oceanographers' slang; why only "biological", what is "update" and "simulation accuracy", why "In reality..", "applicability" to what purposes? Please, reformulate more carefully; 92 – "The reanalysis is mainly based on. . ." Consider, please, replacing something like with "The success of reanalysis. . ." or "The confidence in reanalysis is based on (or stems from). . ." or similar; 94-96 – neither ICES nor SHARK "are monitoring" the Baltic Sea, both just maintain databases with monitoring results, correct appropriately; 104 – in that context a reference to Gustafsson et al. (2012) would be more appropriate in addition to- or instead of Savchuk et al. (2008); 110-111 – is ". . .a better assessment of HISTORICAL changes in the nutrient budgets of the water column and (OS – especially) sediments. . .", true and legitimate aim of this study? Where are historical changes then? 119 – unusual usage of "sea surface heights", replace, please, with "sea level (variations)"; 148 vs. 165 – is it SHARK only or SHARK and BED together? If the later, then there are much more observations in BED, for instance, for the Gulf of Riga; 178-180 vs. 81-82 – repetition, delete, perhaps, from Introduction; 182 – instead of "we focus . . . on nutrient budgets and transports. . .", perhaps, "we

focus . . . on nutrient transports and budgets derived from them. . ." would better reflect both the focus and importance of results; 185-186 – consider simplification as ". . . long- term trends in eutrophication as indicated by Secchi depth (Section 5.4)", because if the water transparency can be used as indicator of the eutrophication as the entire phenomenon, it seems too far-fetched to use it for evaluation of the "excess of nutrients in the water column". 198-199 – what does ". . .positive impact on the model simulation" mean, improved model-data comparability, or model-data resemblance or similar? Is it unexpected? 216 – perhaps, ". . . how data assimilation makes simulated nutrient dynamics in the Baltic proper look more realistic" would be more correct introduction to Fig. 4? 266 – concentrations should be HIGHER not GREATER 268 – Why AMPLITUDES, most common meaning is as the measure of range, fluctuation, difference between maximum and minimum, i.e. large amplitude could mean small NET transports. Maybe, MAGNITUDE? 285 – maybe, "contrast" would be better word than "contradiction"? 306 – What "uptake and deposition of DIP", by which process (es)? 310 – "taken up" or retained? 311-313 – needs better, clearer explanation 315 – Which "vertical exchange", in the water column or along the bottom, how estimated? 380-388 vs. 177-178 – Has not initialization somewhat adjusted the fields? In any way, these considerations once more call for presentation of sediments' pools 428-432 – There is a confusion and misinterpretation about P loads that should be corrected. Possible underestimation of P load was guessed by Savchuk and Wulff (2007) only for the Gulf of Riga. In all other basins, HELCOM data on unfiltered samples were used and GF load of 7 Kt P/yr used by Savchuk and Wulff (2007) are actually very close to the latest compilation by Knuuttila et al. (JMS, 2016). However, the loads in the 1970s and especially, the 1980s were larger indeed. 454 – Isn't location of halocline and, correspondingly, different volumes of hypoxia prone layers a rather important explanation? 484 – Is it denitrification and not PP? Why?

---

## Referee Comment (RC2) · Anonymous Referee #2 · 28 Oct 2016

**General comments**

In this manuscript the authors use a numerical model in combination with data assimilation to estimate nutrient fluxes within the Baltic Sea. They show that the data assimilation scheme greatly improves the results in terms of spatiotemporal concentrations fields. Without data assimilation the model have significant bias in both the annual cycle of the surface layers as well as spatial distribution of nutrient levels, but as shown, the assimilation procedure eliminate significantly of these systematic biases in a very impressive way.

I am unfortunately not at all familiar with data assimilation methods. I tried to get a quick grip on what and how it is done by reading the method description in not only this

manuscript, but also previous papers by the authors. Unfortunately, my background knowledge is too small to really understand even the basics of how it is done. Therefore, I hope that another reviewer is able to penetrate the technicalities of the method and judge its applicability. I can only see the end result and that the assimilated model results really do resemble the reality at the scales presented. I think given that the end results are useful for a wider community and focus on the discussion is not on the technical aspects, it would be useful if the authors include a brief paragraph describing in words how observations and model are merged in the assimilation procedure.

Liu et al presents a solid reanalysis of 4 dimensional nutrient fields in the Baltic Sea. The nice correspondence with observations indicate that resulting data set is probably the best available data set and should provide useful for many purposes. Further that present interesting spatial budgets on both fine and basin-wide scales. One can, of course, question our knowledge of the certainty of the detailed source/sink calculations, but anyway the results are interesting and could definitely be considered best available.

Given the journal one could have wished for deeper analysis of the results in terms of biogeochemical processes. Because of my limited understanding of the methodology I cannot really advice on how far such analysis could go, but now there is very little analysis on whether the spatial fields of sources and sinks may be due to or how they are connected to various processes.

Although discussion is rather weak, I think the results are interesting enough, both in terms of the apparently excellent data quality the method results in as well as the Baltic Sea specific results on nutrient fluxes that I recommend publication.

In general, by relatively small effort, the manuscript text can be improved and I provide some, hopefully helpful, comments below to most sections.

Specific comments

Section 5.1 It is not surprising that the authors find some significant RMSD for e.g.

ammonia in the 1970s. There are substantial temporal trends in data quality and consistent high-quality data is generally achieved only after international inter calibration became standard in the first half of the 1990s. I also believe that ammonia is one of the parameters with largest errors in the 1970s, while phosphate and nitrate was more reliable.

I do not understand "stability" of the assimilation, but that is surely due to my ignorance of the methodology.

Section 5.2 The improvement in capturing the seasonal cycle is impressive. When I study figure 4 in Liu et al (2014) referred to in the text, it seems however, that the improvement is not due to the improved halocline only, but really due to the assimilation of chemical variables. In that figure DIN and DIP seem to be worse when only S and T is assimilated. I am not exactly sure how much interpretation on processes that can be done comparing different assimilated runs, but it seems that when assimilating only S and T, the model fails in using the additional nutrients mixed up. However, I agree that a prerequisite for a deep spring bloom is a deep halocline.

Section 5.3 Also here the improvements are impressive and the spatial variations in winter nutrient concentrations are well captured. This really gives credibility to use these results in flux calculations.

Section 5.4 Secchi depth is a complex variable including strong dependence also on coloured organic matter. It is evident that a higher Secchi depth is obtained using the assimilation, but calculating Secchi depth in the Baltic Sea from modeled algae biomass is not really well constrained so one could argue that by recalculating Secchi using somewhat different attenuation from CDOM could also give a fit to observations with the model without assimilation. Since temporal variation is not captured (which may be due to other causes than biomass), there is no way of knowing which calculation is actually the best and thus applicability of Secchi depth for validation is not very promising. Therefore I suggest that you can remove this section and the associated

**figure.**

Section 5.5 I am not really sure what these horizontal fluxes tell us! Section 5.6 Does the assimilation as such affect conservation or constitute a part of the source/sink? Baring in mind my limited understanding of the methodology, I am wondering whether by having an underlying model simulation with error, corrected by the assimilation scheme the total source/sinks may give some erroneous results? However, I guess if you just integrate currents times concentrations, there should not be any problem.

These results are quite interesting, although a bit challenging to understand. Perhaps it would be somewhat easier to explain if Total P (N) and DIP (DIN) were used instead of Org P (N). The totals would then give the net source/sink of the nutrient and the inorganic show the "gross" source/sink due to net turnover.

It would be easier to read if the comparison with Eilola 2012, was postponed to the discussion. Now, I think the main results from this study is unnecessary difficult to follow, because of the frequent comparison with the previous paper.

Section 5.7 To my knowledge, the model used does only include bio available nutrients. This is fine but should be clearly stated to avoid confusion. Especially for nitrogen, there is a significant net flux through the system of refractory N that is not captured here. I further assume that the budgets are made summing inorganic and organic nutrients, but adding a sentence about that makes it easier for the reader to follow. I am confused by the fact that the budgets in figs 10-11 does not add up. A small net could be attributed to changes in water column storage, but looking for example at phosphorus in Gulf of Finland the net is 8.6+54.7-50.7-6.7 = 12.6 -6.7 = 5.9 kton/yr. This is far too much to be storage change. I thought that it could be that only a part of the load was used, but looking at Gulf of Riga there is a net loss of 1.4 kton/yr. Is it a consequence of the data assimilation? In that case, how should this residual be interpreted? In any case it should be clarified and shown in figures 10-11.

**Discussion**

That gross fluxes are different between approaches are not surprising since it will depend on time-resolution as the authors point out. Oscillating flows due to various processes cause a dispersive transport that to some extent is resolved by the 3D model, but it is not given that the net effect is correct if the processes that regulate the dispersive transport such as e.g., mixing and frontal movements are appropriately modeled. Without really detailed observations of currents and concentrations one have to resort the validation of the dispersive transport to the net effect on e.g. salinity in the basin. Thus, in some sense, the estimate of net transport by a full 3D model may not be that different from the assumptions behind those of using the diagnostic Knudsen approach, i.e. a strong correlation between salinity and the constituent of interest. Having said that, the level of detail is of coarse massively different and the possibilities to make temporal and spatial analyses also greater.

Validation currents and circulation patterns are very difficult and I do not demand that, but it could have been nice with a discussion on how confident we can be in the results of nutrient circulation and source/sink spatial variations in light of how the data assimilation improves circulation. A starting point could be the consequences of that a clear majority of the hydrochemical data has been collected at single locations usually quite central in the basins and not along the stretches of strong circulation. A naive issue that I personally wondering about is whether assimilation of point wise observations may induce spurious circulation patterns?

I would argue that the sub-basin boundaries in the model of Gustafsson etc also (2012) is not arbitrary chosen. As far as possible sub-basin boundaries of this model is chosen according to dynamical constraints such as sills or fronts that can be parametrizised. A discussion of the implications of the high-resolution sink/source fields for our understanding of major processes would have been quite interesting. What does the spatial distribution of e.g. net sedimentation or denitrification imply? What are the pathways for organic matter? I am not sure how far you can take this given methodological limitations, but it could be nice here with a few things and not only referring to other model

simulations.

---

## Author Comment (AC1) · 15 Dec 2016

O.P. Savchuk (Referee)

oleg.savchuk@su.se

**We thank Dr. Savchuk for your very good comments. We have followed all the comments from you and carefully made the improvement in our revision.**

The study deals with application of data assimilation approach to reconstruction of long-term dynamics of 3D nutrient fields as a base for analysis of nutrient transport processes in the Baltic Sea. Both the approach and obtained results are significantly novel in methodological and geographical senses to deserve publishing in "Biogeosciences". However, scientific and presentation qualities should be substantially improved by the major revision of the manuscript along the lines suggested below.

1. General comments and suggestions

1.1 Objectives and applicability. The assimilation of whatever available data is fully justified for an improvement of short-term forecasting of hydrophysical fields aiming at the search-and-rescue operations, propagation and expansion of catastrophic spills as well as management of the maritime activity. However, its applicability for long-term hindcasts of biogeochemical phenomena and properties requires careful consideration and clear explanation of the purposes/objectives of the assimilation (why and what for). Such considerations and explanations should already be given in the Introduction section, with particular attention to the limitations, especially non-conservativeness of the approach (what can and cannot be done).

**Response**: We have specified the aims of data assimilation in the introduction more clearly. The data assimilation meets the gap between observations and numerical modeling in this study. We aim to reproducing the ocean biogeochemical state with the help of information from both observations and a coupled physical-biogeochemical model. The results of the reanalysis can be used to estimate the water quality and ecological state with high spatial and temporal resolution in regions and during periods when no measurements are available. Regional and local model studies may use the data as initial and boundary conditions. Further, nutrient transports across selected cross-sections or between vertical layers might be calculated with high resolution and accuracy taking the complete dynamics of primitive equation models into account. This information cannot be obtained from neither observations alone or from model results without data assimilation because the latter might have large biases in both space and time. We assess the nutrient budgets of the water column and sediments, as well as of the nutrient exchanges between subbasins and between the coastal zone and the open sea. As a reanalysis can never be dynamical consistent and does not preserve mass, momentum and energy (see our response to 1.2), the calculated budgets are compared to the results of other studies to evaluate our results meant as consistency check. Hereby, we follow studies of other regions applying data assimilation for a biogeochemical reanalysis on long-term scale.

For example, Teruzzi et al. 2014. Journal of Geophysical Research, 119, 1–18.

Ciavatta, S., et al. 2016, J. Geophys. Res. Oceans , 121 , 1824–1845.

Fontana, C., et al.,2013, Ocean Sci., 9, 37-56.

In the introduction section we will further clarify the already listed limitations of data assimilation with respect to estimating nutrient budgets and we will rewrite the objectives of this study.

1.2 Artificial non-conservation. Biogeochemical variables are non-conservative by definition, while the entire models of biogeochemical cycles are usually designed as conservative, i.e. explicitly accounting for all the external and internal sources and sinks of the matter. In such models (including the implemented RCO-SCOBI system), the dynamics of simulated nutrient fields is determined by continuous, mutually adjusted interaction of physical transport and biogeochemical transformation processes. If these 4D fields (x, y, z, t) are not absolutely identical to the corresponding fields reconstructed from observations, then an every act of "correction" of simulated towards reconstructed fields during assimilation procedure would create in the model fictitious 3D sinks and sources of the matter not generated by either transport or transformation processes. These fictitious fluxes of nutrients are then included into biogeochemical cycles, thus making the model erroneously non-conservative. Evidently, the studies of eutrophication and biological productivity in general are particularly vulnerable for these effects of data assimilation. As can be deduced, for instance, from Figs. 3-5, such effects are quite substantial.

On the other hand, with a certain confidence in simulated transport agents (water currents and mixing) supported, e.g. by the plausible dynamics of "conservative" salinity (e.g. as in Liu et al. 2013), the "corrected" fields of nutrients could be used for improving simulation of nutrient transport processes. Here, again, the discussion on how such improvement would affect simulation of transformation processes and, in turn, would be affected by them could significantly augment the scientific value of the paper. Also, the questions arises – could not the same results regarding transport processes been achieved just with the "observed" nutrient fields used for assimilation, without running and "jerking/correcting" the biogeochemical model.

In any case, the artificial non-conservativeness should be explicitly acknowledged and explained, its effects evaluated, presented, and discussed, in addition to- and, perhaps, together with analysis of biases by means of RMSD. The estimates of non-conservation and its spatial and temporal dynamics must be computed from a difference between model fields before and after acts of assimilation, starting from the initial conditions.Then the knowledge of needed "correction" can also be used in pinpointing possible deficiencies in the biogeochemical parameterizations.

**Response**: In the long-term simulation, the new initial condition for an assimilation cycle differs from the ending ocean state of the last cycle when at that time observations are available. In this sense, the data assimilation introduces sources and sinks of the nutrient cycles by interrupting the model simulation and adjusting the initial condition. However, we provide the "optimal" initial condition with data assimilation for the RCO-SCOBI for every simulation cycle. It means we don't change the equations of the RCO-SCOBI and just integrate currents and concentrations. The simulation process is conservative during the simulation between two assimilation occasions.

We agree with Dr. Savchuk that the data assimilation affects conservation properties for the long simulation as a whole. Although the reanalysis is conserved during every "independent" simulation cycle, the adjustment of data assimilation implicitly creates unknown complementary sources or sinks to the biogeochemical model. The magnitude of these adjustments depends on the bias between model and observations. The artificial sources/sinks are directly related to the model biases. Figure 3 shows that the model has large biases during the beginning of the simulation. However, data assimilation has corrected the mismatch between model state and observation to an "optimal" level during an initial adjustment period. After the adjustment period, the mismatch between model and observation becomes small and the successive adjustment due to data assimilation also becomes small (Liu et al. 2014). Further, the adjustment of data assimilation is related to the spatial-temporal coverage of observations. Here we assimilated only observed profiles into the model.

The advantage of the data assimilation is that model variables at any station are very likely more accurate than the model output without data assimilation. For instance, time series of profiles or

transports across vertical sections have very likely a smaller bias compared to observations than the corresponding model results without data assimilation. Compared to available observations the information from the model is higher resolved and homogeneous in space and time. Of course, it is difficult to evaluate the quality of model results at high resolution because independent observational data sets are usually missing. An exceptional effort to utilize independent data was done by Liu et al. (2014) showing that the statement about the added value of data assimilation is true for the available, independent cruise data at high resolution. However, one can not expect that budgets calculated from the summation of fluxes from model results with data assimilation are more accurate because usually small artificial sources and sinks from the data assimilation are becoming as important as physically motivated sources and sinks when sums of fluxes are compared. Hence, we calculated budgets with the aim to evaluate the reanalysis data and to estimate the magnitude of artificial sources and sinks by comparing our results with other studies using only observations. We are aware that it is impossible to claim that our budgets are more accurate than those budgets that are derived from observations only despite the higher temporal and spatial resolution in model outputs. Hence, the advantage of the reanalysis is that measurements are extrapolated in space and time based upon physical principles of the model. However, the disadvantage is that the reanalysis data does not obey conservation principles. We will discuss advantages and disadvantages of the reanalysis in more detail in the revised version of the manuscript.

1.3 Plausibility of the RCO-SCOBI model. The RCO-SCOBI model has been extensively used for forecasts (aka projections) of possible changes in the Baltic Sea biogeochemistry under different scenarios of driving forces, practically by the same authors. Therefore, the scientific value of the paper could be significantly increased by the discussion and speculations on how the model's deficiencies in simulation of transport flows and transformation fluxes, which are revealed due to the data assimilation, for instance, in the form of RMSD, could affect the predictions. Good starting point could be a statement at line 387.

**Response:** RCO-SCOBI has been widely used for the Baltic Sea and the model was carefully evaluated using various observational data sets. As any other model RCO-SCOBI had to be calibrated because many processes including sources and sinks of nutrients are not detailed enough known. Hence, an "optimal" parameterization of unresolved processes is one of the requirements for the predictive capacity of the model. Further requirements to calculate correct transports and transformation processes in addition to optimized model equations are high-quality atmospheric and riverine forcing data, and high-quality initial and lateral boundary conditions.

We discussed already in the present version of the manuscript why FREE has so large biases compared to the results by Liu et al (2014, Tellus A) and compared to biogeochemical observations. Most of the large differences are caused by imperfect initial conditions, which can be seen from the temporal evolution of the RMSD (Figure 3).

For projections of future climate and for nutrient load abatement scenarios the reanalysis has a very high scientific value as reference data set for the historical period of the climate simulations. The evaluation of the regionalized climate (the statistics of mesoscale variability, e.g. the mean state) during the historical period can be done much more accurate based upon the reanalysis data than with sparse observational data. For instance, it is very difficult to calculate the climatological mean state just from observations that are casted only during the ice-free season of the year. Using a reanalysis as reference data for historical climate is a common method in regional climate studies of the atmosphere. Here we provide a corresponding data set for the ocean to evaluate simulated present-day climate. We will add a paragraph to the discussion to highlight the value of reanalysis data sets for climate studies.

1.4. Description and explanation of Methods. All the methods implemented in the manuscript must be described in more detail and, considering an intended expansion of the paper's coverage from the "hydrophysical" audience over the "Bio-Geo-Chemical" one, in somewhat more popular style. Assimilation procedure. In addition to references to (Liu et al. 2013, 2014), several details, especially those important for magnitude and distribution of 4D fictitious fluxes, must be repeated and explicitly explained in this paper as well. The explanations should include, for instance, such details as: a) verbal description of procedure for reconstruction of "observed" fields used further in assimilation and in calculation of RMSD in FREE and REAN experiments, b) spatially and temporally varying uncertainties of such fields determined by the scarcity and sparsity of observations, c) frequency of the assimilation acts and its possible effects on the difference between model and observation used in calculation of RSMD (Liu et al., 2014), and whatever else would be necessary for further presentation and discussion of issues from Comment 1.2 above. Without such clarifications, three sentences at lines 170-173 look as isolated abracadabra and might seem almost useless.

Nutrient transports, trends, and budgets. The exact definitions of all the nutrient transports, trends, and budgets measures and characteristics together with algorithms of their calculation, including derived units, should be clearly presented already in Methods. This will clarify possible confusions with the usage and interpretation of the terms vs. phenomena, commented in details below, in Section 2.

**Response**: We will detail and rewrite the text in the method's description according to your comments. See the sections 4 "Methodology and Experimental Setup".

2. Specific comments and suggestions.

2.1 "Cycling" in the title and similar statements to that effect elsewhere Accordingly to comments 1.1-2 above, the non-conservative model cannot be used for comprehensive studies of nutrient CYCLING. Hence, the title should be modified – consider, please, something like "Nutrient TRANSPORTS in the Baltic :::" instead. Correspondingly, the usage of "cycling" and similar statements and expressions about transformation processes should be carefully revaluated throughout the entire text, for instance, at lines 80, 189, 310, 306-307, 362-363, 466, and throughout the entire Section 5.6,

**Response**: Following your suggestions, we will change the text and use nutrient transports instead of nutrient cycles.

2.2 Calculation of RMSD. Line 194 – What is the meaning of "overall" and "monthly mean" in "the overall monthly mean RMSDs" and how they were calculated – for how many fields per month? covering the entire Baltic? cell by cell for interpolated "observational" fields or only for cells with the real observations?

**Response**: We add the following Equation to specify the calculation process of RMSD in the revised manuscript.

The overall monthly mean RMSD is calculated by the following formula:

$$RMSD = \frac{1}{N_j} \sum_{j=1}^{N_j} \sqrt{\frac{1}{N_t} \sum_{i=1}^{N_t} (\varepsilon_t^i)^2}$$

where $N_t$ is the number of the observations at assimilation time $t$ and $N_j$ are the number of days observed in one month for one field for entire Baltic Sea. $\varepsilon_t^i = x_{sim}^i(t) - x_{obs}^i(t)$ represents the model-observation difference at the time t at the $i^{th}$ observation position. $x_{sim}$ and $x_{obs}$ are the modeled and observed field. We calculated $\varepsilon_t$ at only the observation position at the time $t$, which is calculated by mapping the corresponding model field to the observation space.

2.3 Nutrient transports. Explain and clarify, please, involved terms and interpretations – What does "net" (which is usually used with the word "exchange" and represents a difference between inputs/imports and outputs/exports) mean at lines 17, 259-260, 277, 300, 338, 356, 360, and 492; – Why some characteristics related to single grid cells or a grid "column" are called "net", has it something to do with the difference between in- and out- transport flows or/and is it meant to account for local changes due to transformations, causing difference between inflows to the cell (column) and outflows from it? For instance, at line 694 – How exactly the vertical averages and vertical integrals (e.g. line 259) have been computed? Why ANNUAL average is expressed in ton/ km/MONTH (Fig. 7, lines 694-697)? Would vertical averages multiplied by the depth of grid point be equal to vertical integrals? What is the point presenting/contrasting/comparing (e.g. in Fig. 7) vertically averaged transport for the locations with, for instance, 200 and 20 m depths? – Definitions and explanations for calculations of nutrient sources and sinks from integral transports would be helpful in understanding and interpretation of Section 5.6. Some consideration and discussion on how much the sinks and sources could depend on which transformation processes and how much they would be determined by fictitious fluxes might be useful too. Also, check the consistency of term's usage both in the text and, especially in legend to Figs. 8 and 9 (annual average IMPORT (transports?); again ANNUAL is expressed on per MONTH basis.

**Response:** We add the following equation to explain the calculation process of the nutrient transports in every grid 'column' or 'cell'. The vertically averaged transport ($VA_{Trans}$ ) at every horizontal grid section at a simulation time is calculated with the following formula:

$$VA_{Trans} = \frac{1}{N_z} \sum_{j=1}^{N_z} \iint \rho_f u \, dx \, dz ,$$

where $\rho_f, u, N_z, dx$ and $dz$ are the field concentrations, the current velocity normal to cross-sectional area, the number of wet grid cells in one water column, the horizontal and vertical dimensions of a grid cell (m), respectively.

Here the net transports express the difference between inflow and outflow transports. Both "net" and "exchange" are common usage in the description of transport. Just like you mention here the "net" denotes the difference between inputs/imports and outputs/exports. We will define "net" in the method part of the revised manuscript.

For example. Eilola et al . Ambio., 41, 574–585, 2012. Treguier et al.,Ocean Sci., 10, 243–255, 2014.

The "net" usages also denote the local transport change in every section or grid cell or grid 'column'.

We change the "ANNUAL average" to "Monthly average" in the corresponding text. The calculated process referred to the above Equation.

The vertically averaged nutrient transports present the direction and magnitude of the nutrient transports in every water "column" in the Baltic Sea. The Figure 7 shows the mean net horizontal nutrient transport in each cell of the horizontal model grid. From that we can get the distribution of the direction and magnitude of nutriment horizontal transport in the Baltic. For

example, the magnitudes of DIP/DIN transport are stronger in the east Gotland basin than that in Gulf of Finland.

Definitions and explanations of sources and sinks have been given in the text of Section 5.6(also see our response to 2.6). Further, we give how transport is calculated in every grid cell or 'column' (see Equation in the reply to 2.4 ).

We changed the legend usage in the Figure 9. And we also clarify the description of net flow in the Section 5.6, which use the consistent term's usage description.

Actually, if we use the Knudsen approach (exactly as described in Savchuk 2005) to calculate the water flow transports we obtain results similar to Savchuk (2005) (see figure below).

[Figure]

Figure.  Water flows between the Baltic Sea basins (km$^3$ year$^{-1}$ ) averaged over 1970-1999. External water inputs are the sum of net freshwater supply from river runoff and from atmosphere.

2.4 Nutrient budgets. Explain, please, how the budgets were computed: – How nutrient in- and outflows (as product of velocity and concentration) been obtained from integrals of continuous computations for period 1970-1999 or from averaging of monthly or annual integrals?  – How have annual sink/sources been calculated?  Have the transformation processes (sediment-water exchanges, burial, nitrogen fixation, denitrification) been accounted for? – How trends in Table 1 been estimated?  What does P sources in the KT, GF, and BB (sic!) as well as N source in GF mean? – How the total amounts (pools) of nutrients were calculated, by averaging of which fields, integrated with which frequency?

**Response**: The calculations of nutrient budgets will be better explained in the revised version. The nutrient flow for the budgets is calculated by the similar method to the above shown integral equation at the selected borders of Baltic subbasins. We obtained the annual average nutrient flow from integrals of continuous computations for period 1970-1999.

In the nutrient budgets the P and N external sources are computed from the combined supplies from land and atmosphere. Nitrogen fixation is not included in the external supplies. The sediment sinks are calculated from the difference between the net deposition of nutrients to the sediments and the release of nutrients from the sediments.

The model includes all these transformation processes (sediment-water exchanges, burial, nitrogen fixation, denitrification). The results have taken these processes into account. (refer to Eilola et al,  J. Mar. Syst., 75, 163–184, 2009 and Almroth-Rosell et al, Journal of Marine Systems, 144, 127–141, 2015.)

The potential impact from artificial sources or sinks due to data assimilation is of course also included in the results. Because of the unknown impact from this "process" it is better to avoid detailed discussions especially about the changes in the nutrient pools. The trends in Table 1 are

calculated from the differences between the nutrient inputs and nutrient exports seen in Figures 9 and 10.

The total amounts (pools) of nutrients were calculated as the sum of the inorganic and organic nutrients in the water. The total amounts of nutrients for every grid cell were calculated with the averaged nutrient concentration of corresponding grid cell during the period 1970-1999 and the formula:

$$Total = \iint \rho_f dx dz,$$

where $\rho_f, dx$ and $dz$ are the field concentrations (including nutrients from phytoplankton, zooplankton, detritus and dissolved nutrient) the horizontal and vertical dimensions of a grid cell (m), respectively. And then total amounts of nutrients are the sum of the nutrients of all water grid "cell" in every subbasins.

These explanations are necessary but not sufficient for understanding how 30-year average annual "tendencies" (trends? deviations?) agree with pools? Most illustrative are P sources. In BB, 0.8 Kt P/yr *30 yrs=24 Kt P comparing to the pool of 5.9 Kt P; in GF, 5.9 Kt P/yr *30 yrs=177 Kt P comparing to the pool of 29.9 Kt P. Where has such hefty P excess gone, accumulated in the sediments? Evidently, the changes of nutrient pools in sediments must be included into consideration as well regardless of how plausible they are.

**Response**: We redefine the borders of the subbasins (Fig. 1) and recalculate the total nutrient budget based on the new borders. Meanwhile we corrected the mistake caused by the unit transform. The results are regarded reliable and reasonable. For example, the net phosphorus tendency for the Gulf of Finland is 24.3-22.5+8.6-6.7 = 3.7 Kton/yr. Further, in the Bothnian Bay, the net nitrogen tendency is zero. Comparison with the results of Savchuk (2005, 2007) based on Knudsen approach, the difference is mainly caused by the external supply from atmosphere and land. But phosphorus tendency in Gulf of Riga still a net loss of 0.5 Kton/yr. The difference between our result and Savchuk (2005) is due to different internal removal. Our results and Savchuk (2005, 2007) are treating different periods, the loads in the 1970s and the 1980s were larger indeed compared the loads in 1990s.

– Legend to Figs. 10 and 11 says: "External nutrient inputs are separated into terrestrial and atmospheric sources. Terrestrial loads are reduced by phosphorus retentions for the coastal zones." However, external inputs are presented with single numbers. Is it a sum of terrestrial and atmospheric loads, then the word is "combined"? What is the coastal P retention, how it was estimated and which values were prescribed? Was N inputs treated in a similar way?

**Response**: the number of external inputs is a sum of the supply from atmosphere and land. We change the word used in these figures description. We remove the text "Terrestrial loads are reduced by phosphorus retentions for the coastal zones" since our model has consider these process during the model calculating nutrient flux.

Similar explanations and considerations, starting from algorithm of calculation should be given also to horizontally integrated flows at transects (Fig.12, lines 349-378) with special attention paid to explanation of the purpose of their analysis in a view of complex picture of water circulation and nutrient transports in Fig. 7. Considerations about possible contributions of transformation vs. fictitious processes would be appropriate in Section 5.7 or in discussion of presented results as well.

**Response**: we have given the answer for these comments. Please refer to the reply to 1.2 and 2.3.

2.5 Secchi depth (see also comment for lines 185-186 below). The water transparency seasonal variations and long-term trends depend on too many factors that either are not included in the model (e.g. CDOM and SPM distribution and variation) or are determined by complicated feedbacks from transformation processes (e.g. primary production and sedimentation of decomposing organic matter) to be used as unequivocal indicator of improved simulation of the nutrient fields. In result, the related analysis (lines 250-253) looks weak and unconvincing, for instance, the decrease of inorganic nutrients should cause the decreased primary production and how realistic is that? Or is it a correct effect by the wrong reason? Therefore, I would recommend deleting consideration of Secchi depth from the paper entirely. However, if the authors will chose to retain these considerations then a few words about how Secchi depth is estimated in the model (what it does and does not account for) would be useful for readers.

**Response**: we followed the suggestion by the reviewer and deleted this section about the Secchi depth in the revised manuscript.

2.6 Presentation of pelagic and sediments pools. As it appears from Comments 2.4 and lines 380-388 in Discussion, presentation of pelagic and sediment nutrient pools could help to untangle several issues in interpretation of results

**Response:** As mentioned earlier, the potential impact from artificial sources or sinks due to data assimilation is of course also included in the results. Because of the unknown impact from this "process" it is better to avoid detailed discussions especially about the changes in the nutrient pools.

3. Minor things, technical corrections and language cosmetics.

We will in the revised version have several major changes in the text that may affect the interpretation of the detailed suggestions given by the reviewer. We will seriously consider and take into consideration all minor comments from the reviewer also in the reworking of the text.

Lines: 3 – I guess, it is Eilola not Eolila;

**Response:** we correct it in revised manuscript.

11-12 – What is "improvement in ::: concentrations"? Consider, please, something like "improved simulation/reproduction/imitation of concentrations" or similar;

**Response**: We change it to "…improved simulation of both oxygen and nutrient concentrations"

33-34 – Perhaps, not as much "living conditions" as redox dependent biogeochemical processes; here the reference to (Conley et al., 2009) or/and (Savchuk, 2010) would be appropriate in addition to- or instead of (Fu, 2013)

**Response**: We change this sentence to "MBIs can significantly affect the biogeochemical processes in the deep basins because of the inflow of large volumes of oxygen-rich water into the Baltic Sea (e.g. Conley et al. 2009; Savchuk, 2010)."

50-54 – poor choice of words: " ::: of BIOLOGICAL formulations (either empirical or mechanistic) to UPDATE biogeochemical concentrations" that sounds as (physical) oceanographers' slang; why only "biological", what is "update" and "simulation accuracy", why "In reality..", "applicability" to what purposes? Please, reformulate more carefully;

**Response**: We rewrite it in revised manuscript. To clarify, now we delete "In general, coupled physical-biogeochemical models use a variety of biological formulations (either empirical or mechanistic) to update biogeochemical concentrations. As a result, the model formulation and the reliability of their parameterizations play a key role in determining the simulation accuracy of biogeochemical processes. In reality these processes governing the interactions between biogeochemical compartments vary in space and time (Losa et al., 2004; Doney, 1999)." in the revised version.

92 – "The reanalysis is mainly based on ::: " Consider, please, replacing something like with "The success of reanalysis ::: " or "The confidence in reanalysis is based on (or stems from) ::: " or similar;

**Response**: We change it to ":::The success of reanalysis is mainly based on a reliable model:::"

94-96 – neither ICES nor SHARK "are monitoring" the Baltic Sea, both just maintain databases with monitoring results, correct appropriately;

**Response**: We change it to "For example, the International Council for the Exploration of the Sea (ICES) (http://www.ices.dk) and the Swedish Oceanographic Data Centre (SHARK) (http://sharkweb.smhi.se) are collecting the observations with the aim to monitor the Baltic Sea. Furthermore, the Baltic Sea Operational Oceanographic System (BOOS) (http://www.boos.org/) is providing near real-time observations."

104 – in that context a reference to Gustafsson et al. (2012) would be more appropriate in addition to- or instead of Savchuk et al. (2008);

**Response**: We replace the reference to Savchuk et al. (2008) by Gustafsson et al. (2012).

110-111 – is " ::: a better assessment of HISTORICAL changes in the nutrient budgets of the water column and (OS – especially) sediments ::: " , true and legitimate aim of this study? Where are historical changes then?

**Response**: we change description of the aim of this study. Please see the reply to 1.1.

119 – unusual usage of "sea surface heights", replace, please, with "sea level (variations)";

**Response**: we replace the "sea surface heights" by "sea level elevation"

148 vs. 165 – is it SHARK only or SHARK and BED together? If the later, then there are much more observations in BED, for instance, for the Gulf of Riga;

**Response**: Yes, data from SHARK are assimilated into RCO-SCOBI. But data from both SHARK and BED are used for validation. We correct it in revised manuscript.

178-180 vs. 81-82 – repetition, delete, perhaps, from Introduction;

**Response**: we delete the "However, in Liu et al. (2014), only a shorter assimilation experiment for a 10-year period is presented, and so far the stability of the assimilation scheme in multi-decadal simulations has not been shown." in introduction section.

182 – instead of "we focus ::: on nutrient budgets and transports ::: ", perhaps, "we ::: on nutrient transports and budgets derived from them ::: " would better reflect both the focus and importance of results;

**Response**: we accept your comments and change it in the revised manuscript.

185-186 – consider simplification as " ::: long- term trends in eutrophication as indicated by Secchi depth (Section 5.4)", because if the water transparency can be used as indicator of the eutrophication as the entire phenomenon, it seems too far-fetched to use it for evaluation of the "excess of nutrients in the water column".

**Response**: we delete this sentence: "and long-term trends in eutrophication (excess of nutrients in the water column) as indicated by Secchi depth (Section 5.4)".

198-199 – what does " ::: positive impact on the model simulation" mean, improved model-data comparability, or model-data resemblance or similar? Is it unexpected?

**Response**: the positive impact means reanalysis results closer data relative to FREE, which reduce the uncertainly of model simulation.

216 – perhaps, " ::: how data assimilation makes simulated nutrient dynamics in the Baltic proper look more realistic" would be more correct introduction to Fig. 4?

**Response**: we change it according to your comment.

266 – concentrations should be HIGHER not GREATER.

**Response**: We change the word "greater" to "higher".

268 – Why AMPLITUDES, most common meaning is as the measure of range, fluctuation, difference between maximum and minimum, i.e. large amplitude could mean small NET transports. Maybe, MAGNITUDE?

**Response**: We change the word "amplitude" to "magnitude". Thanks for your kind comment.

285 – maybe, "contrast" would be better word than "contradiction"?

**Response**: We change the word "contradiction" to "contrast".

306 – What "uptake and deposition of DIP", by which process (es)?

**Response**: We change this sentence by "This result might be explained by local processes causing the phytoplankton uptake and sediment deposition of DIP.".

310 – "taken up" or retained?

**Response**: it should be "retained".

311-313 – needs better, clearer explanation.

**Response:** The phosphorus sink may also be partly caused by oxygen dependent water–sediment fluxes that bind DIP to ironbound phosphorus in oxic sediments (Almroth et al., 2015). This effect is not included in the Eilola et al. (2012), but might potentially be accounted for by the adjusted DIP transports in REANA. The results of REANA indicate that there is an additional sink but the relative importance of different processes causing this sink (data assimilation or sediment processes) is, however, not possible to evaluate from the reanalysis data set.

315 – Which "vertical exchange", in the water column or along the bottom, how estimated?

**Response**: the "vertical exchange profile" description is related to the internal nutrient sink/source at different water depth (Figure 8). But for clarification, we delete "vertical" in the revised manuscript.

380-388 vs. 177-178 – Has not initialization somewhat adjusted the fields? In any way, these

considerations once more call for presentation of sediments' pools.

**Response**: Both REANA and FREE take the start initial condition from the same earlier run. However, to REANA, we firstly use the data assimilation method to "optimize" the initial condition and then forward the integration. FREE forward the integration based on the non-"optimal" the initial condition.

428-432 – There is a confusion and misinterpretation about P loads that should be corrected. Possible underestimation of P load was guessed by Savchuk and Wulff (2007) only for the Gulf

of Riga. In all other basins, HELCOM data on unfiltered samples were used and GF load of 7 Kt P/yr used by Savchuk and Wulff (2007) are actually very close to the latest compilation by Knuuttila et al. (JMS, 2016). However, the loads in the 1970s and especially, the 1980s were larger indeed.

**Response**: we clarify it by delete this sentence: "However, their total phosphorus load, for example to the Gulf of Finland, is underestimated because the particulate phosphorus fraction is neglected (Savchuk et al., 2012)."

454 – Isn't location of halocline and, correspondingly, different volumes of hypoxia prone layers a rather important explanation?

**Response**: Yes, we also think it is good explanation of model biases. We add it into revised manuscript.

484 – Is it denitrification and not PP? Why?

**Response**: Thank you for the comment. The high productivity in the shallow areas effectively transfers DIN to OrgN. The denitrification act on larger scales and decrease the exports of nitrogen from coastal areas to the deeper areas. The potential impact from artificial sources or sinks due to data assimilation is also included in the results. The discussion in the manuscript will be revised accordingly.

---

## Author Comment (AC2) · 15 Dec 2016

**We thank you for your most helpful and thoughtful comments in the evaluation of our manuscript.**

General comments

this manuscript the authors use a numerical model in combination with data assimilation to estimate nutrient fluxes within the Baltic Sea. They show that the data assimilation scheme greatly improves the results in terms of spatiotemporal concentrations fields. Without data assimilation the model have significant bias in both the annual cycle of the surface layers as well as spatial distribution of nutrient levels, but as shown, the assimilation procedure eliminate significantly of these systematic biases in a very impressive way. I am unfortunately not at all familiar with data assimilation methods. I tried to get a quick grip on what and how it is done by reading the method description in not only this manuscript, but also previous papers by the authors. Unfortunately, my background knowledge is too small to really understand even the basics of how it is done. Therefore, I hope that another reviewer is able to penetrate the technicalities of the method and judge its applicability. I can only see the end result and that the assimilated model results really do resemble the reality at the scales presented. I think given that the end results are useful for a wider community and focus on the discussion is not on the technical aspects, it would be useful if the authors include a brief paragraph describing in words how observations and model are merged in the assimilation procedure. Liu et al presents a solid reanalysis of 4 dimensional nutrient fields in the Baltic Sea. The nice correspondence with observations indicate that resulting data set is probably the best available data set and should provide useful for many purposes. Further that present interesting spatial budgets on both fine and basin-wide scales. One can, of course, question our knowledge of the certainty of the detailed source/sink calculations, but anyway the results are interesting and could definitely be considered best available. Given the journal one could have wished for deeper analysis of the results in terms of biogeochemical processes. Because of my limited understanding of the methodology I cannot really advice on how far such analysis could go, but now there is very little analysis on whether the spatial fields of sources and sinks may be due to or how they are connected to various processes. Although discussion is rather weak, I think the results are interesting enough, both in terms of the apparently excellent data quality the method results in as well as the Baltic Sea specific results on nutrient fluxes that I recommend publication.

**Response:** We detail and rewrite the text in the method's description according to your comments. See the sections 4 "Methodology and Experimental Setup", which describes how the observations and model are merged in the assimilation procedure.

In general, by relatively small effort, the manuscript text can be improved and I provide some, hopefully helpful, comments below to most sections.

Specific comments

Section 5.1 It is not surprising that the authors find some significant RMSD for e.g. ammonia in the 1970s. There are substantial temporal trends in data quality and consistent high-quality data is generally achieved only after international inter calibration became standard in the first half of the 1990s. I also believe that ammonia is one of the parameters with largest errors in the 1970s, while phosphate and nitrate was more reliable.

I do not understand "stability" of the assimilation, but that is surely due to my ignorance of the methodology.

**Response:** Thanks for specifying the quality of the ammonia observation.

Here we mean the assimilation results give a reliable estimation of the ocean state during the whole period. EnOI relies on the selected ensemble sample to estimate the background error covariance of model. The poor sample ensemble can cause the failure of the analysis. With the evolution of simulation, the performance of the data assimilation is different. The success of data assimilation at one time can't guarantee continued success of data assimilation at another time. Therefore, the "reliable" of a data assimilation system is key to the reanalysis. The RMSDs in Figure 3 denoted our estimation with EnOI is successful during the whole simulation period, which proved that our data assimilation system is valid and "reliable". To clarify, we instead "stability" by "reliable" in revised manuscript.

Section 5.2 The improvement in capturing the seasonal cycle is impressive. When I study figure 4 in Liu et al (2014) referred to in the text, it seems however, that the improvement is not due to the improved halocline only, but really due to the assimilation of chemical variables. In that figure DIN and DIP seem to be worse when only S and T is assimilated. I am not exactly sure how much interpretation on processes that can be done comparing different assimilated runs, but it seems that when assimilating only S and T, the model fails in using the additional In nutrients mixed up. However, I agree that a prerequisite for a deep spring bloom is a deep halocline.

**Response:** As shown by Liu et al. (2014), adjusting the physical condition for biogeochemical model doesn't guarantee the better biogeochemical simulation.

Requirements to calculate correct simulation in additional to optimized model equations are high-quality atmospheric and riverine forcing data, and high-quality initial and lateral boundary conditions. As any other model, RCO-SCOBI had to be calibrated because many processes including sources and sinks of nutrients are not detailed enough known. Hence, an "optimal" parameterization of unresolved processes is one of the requirements for the predictive capacity of the model. The "optimal" physical forcing field is one of conditions to guarantee the correct the biogeochemical simulation. Assimilating only S/T will possibly break the balance of physical-biogeochemical condition, which provides the "optimal" initial condition for the circulation model and maybe degrade the usage of the former "optimal" parameterization for biogeochemical model. As a result, the physical-biogeochemical simulation using only T/S assimilation is done with "non-optimal" initial condition. Therefore, both physical and biogeochemical observations are necessary to be assimilated into the model to produce the "optimal" initial condition for a coupled physical-biogeochemical model simulation.

Section 5.3 Also here the improvements are impressive and the spatial variations in winter nutrient concentrations are well captured. This really gives credibility to use these results in flux calculations.

**Response:** thanks for your comments!

Section 5.4 Secchi depth is a complex variable including strong dependence also on coloured organic matter. It is evident that a higher Secchi depth is obtained using the assimilation, but calculating Secchi depth in the Baltic Sea from modeled algae biomass is not really well constrained so one could argue that by recalculating Secchi using somewhat different attenuation from CDOM could also give a fit to observations with the model without assimilation. Since temporal variation is not captured (which may be due to other causes than biomass), there is no way of knowing which calculation is actually the best and thus applicability of Secchi depth for validation is not very promising. Therefore I suggest that you can remove this section and the associated

**Response:** Following the advice of both reviewers we delete this content from the revised manuscript.

Section 5.5 I am not really sure what these horizontal fluxes tell us!

**Response:** The aims of presenting mean horizontal nutrient currents in the Baltic Sea is helpful to address the description of the nutrient exchanges between subbasins and between the coastal zone and the open sea in manuscript. The nutrient transport in Baltic Sea is differing from other regions because of its physical and biological condition (e.g. the shallow mean water depth, much river runoff, the weak tide, the much source/sink). The horizontal distribution of the nutrient transport gives the hint to detect the intensity and direction of the nutrient transport.

Section 5.6 Does the assimilation as such affect conservation or constitute a part of the source/sink? Baring in mind my limited understanding of the methodology, I am wondering whether by having an underlying model simulation with error, corrected by the assimilation scheme the total source/sinks may give some erroneous results? However, I guess if you just integrate currents times concentrations, there should not be any problem. These results are quite interesting, although a bit challenging to understand. Perhaps it would be somewhat easier to explain if Total P (N) and DIP (DIN) were used instead of Org P (N). The totals would then give the net source/sink of the nutrient and the inorganic show the "gross" source/sink due to net turnover. It would be easier to read if the comparison with Eilola 2012, was postponed to the discussion. Now, I think the main results from this study is unnecessary difficult to follow, because of the frequent comparison with the previous paper.

**Response**: In the long-term simulation, the new initial condition for an assimilation cycle differs from the ending ocean state of the last cycle when at that time observations are available. In this sense, the data assimilation introduces sources and sinks of the nutrient cycles by interrupting the model simulation and adjusting the initial condition. However, we provide the "optimal" initial condition with data assimilation for the RCO-SCOBI for every simulation cycle. It means we don't change the equations of the RCO-SCOBI and just integrate currents and concentrations. The simulation process is conservative during the simulation between two assimilation occasions.

We agree that the data assimilation affects conservation properties for the long simulation as a whole. Although the reanalysis is conserved during every "independent" simulation cycle, the adjustment of data assimilation implicitly creates unknown complementary sources or sinks to the biogeochemical model. The magnitude of these adjustments depends on the bias between model and observations. The artificial sources/sinks are directly related to the model biases. Figure 3 shows that the model has large biases during the beginning of the simulation. However, data assimilation has corrected the mismatch between model state and observation to an "optimal" level during an initial adjustment period. After the adjustment period, the mismatch between model and observation becomes small and the successive adjustment due to data assimilation also becomes small (Liu et al. 2014). Further, the adjustment of data assimilation is related to the spatial-temporal coverage of observations. Here we assimilated only observed profiles into the model.

We want to keep the discussion of internal dynamics of inorganic and organic nutrient. As mentioned earlier, the potential impact from artificial sources or sinks due to data assimilation is included in the reanalysis results. Because of the unknown impact from this "process" it is better to avoid detailed discussions about the net sources and sinks.

We move the comparison with Eilola et al. (2012) to the discussion section.

Section 5.7 To my knowledge, the model used does only include bio available nutrients. This is fine but should be clearly stated to avoid confusion. Especially for nitrogen, there is a significant net flux through the system of refractory N that is not captured here. I further assume that the budgets are made summing inorganic and organic nutrients, but adding a

sentence about that makes it easier for the reader to follow. I am confused by the fact that the budgets in figs 10-11 does not add up. A small net could be attributed to changes in water column storage, but looking for example at phosphorus in Gulf of Finland the net is 8.6+54.7-50.7-6.7 = 12.6 -6.7 = 5.9 kton/yr. This is far too much to be storage change. I thought that it could be that only a part of the load was used, but looking at Gulf of Riga there is a net loss of 1.4 kton/yr. Is it a consequence of the data assimilation? In that case, how should this residual be interpreted? In any case it should be clarified and shown in figures 10-11. That gross fluxes are different between approaches are not surprising since it will depend on time-resolution as the authors point out. Oscillating flows due to various processes cause a dispersive transport that to some extent is resolved by the 3D model, but it is not given that the net effect is correct if the processes that regulate the dispersive transport such as e.g., mixing and frontal movements are appropriately modeled. Without really detailed observations of currents and concentrations one have to resort the validation of the dispersive transport to the net effect on e.g. salinity in the basin. Thus, in some sense, the estimate of net transport by a full 3D model may not be that different from the assumptions behind those of using the diagnostic Knudsen approach, i.e. a strong correlation between salinity and the constituent of interest. Having said that, the level of detail is of coarse massively different and the possibilities to make temporal and spatial analyses also greater.

Validation currents and circulation patterns are very difficult and I do not demand that, but it could have been nice with a discussion on how confident we can be in the results of nutrient circulation and source/sink spatial variations in light of how the data assimilation improves circulation. A starting point could be the consequences of that a clear majority of the hydrochemical data has been collected at single locations usually quite central in the basins and not along the stretches of strong circulation. A naive issue that I personally wondering about is whether assimilation of point wise observations may induce spurious circulation patterns?

**Response:** Thanks for your comments. Yes, the budgets are made summing inorganic and organic bio available nutrients. We add text for clarifying the total nutrient in this section in revised manuscript.

The budget calculation is recalculated with new borders. The results are reliable and reasonable.

Meanwhile we corrected the mistake caused by the unit transform. The results are regarded reliable and reasonable. For example, the net phosphorus tendency for the Gulf of Finland is 24.3-22.5+8.6-6.7 = 3.7 Kton/yr. Further, in the Bothnian Bay, the net nitrogen tendency is zero. Comparison with the results of Savchuk (2005, 2007) based on Knudsen approach, the difference is mainly caused by the external supply from atmosphere and land. But phosphorus tendency in Gulf of Riga still a net loss of 0.5 Kton/yr. The difference between our result and Savchuk (2005) is due to different internal removal. Our results and Savchuk (2005, 2007) are treating different periods, the loads in the 1970s and the 1980s were larger indeed compared the loads in 1990s.

In the Baltic Sea, the mean water depth amounts to about 54m. Mainly wind forcing and topography are the factors that affect the variability of the circulation In the shallow region in the Baltic Sea, where stratification is weak, the surface circulation may affect the sea floor. Further, the topography of the sea floor plays an important role in constraining the circulation and much of the abyssal flow is funneled through passages such as the Denmark Straight. Our reanalysis changes salinity and temperature of seawater but it does not change the horizontal circulation explicitly. Further, we change the stratification in the Baltic Sea which will affect the vertical circulation in our assimilation experiment (Liu et al. 2013). Fu et al. (2011) has validated the improvement of sea level in assimilating temperature and salinity with EnOI method. In this study, we don't change the forcing. With a high-resolution circulation model, physical state variables include the sea level, temperature and salinity. We consider the impact of barotropic and baroclinic balance during the assimilation. Further, Wenzel et al. (2001) proved that, when sea level is assimilated in the circulation model in addition to temperature and salinity to adjust the small-scale variability, the large-scale circulation will not be degraded.

We estimated the assimilation increment according to optimal statistics of the water column in every grid point. The water mass is mainly controlled by the temperature and salinity. We estimated the "optimal" characteristics (temperature and salinity) of water mass in our reanalysis. The "optimal" characteristics will produce the "optimal" hydrological dynamic balance based on the model dynamic equations. As a result, we don't degrade the estimation of horizontal transport

M. Wenzel et al. (2001) Progress in Oceanography 48 73–119.

Actually, we used the Knudsen approach to calculate the water flow transports. We obtained results similar to Savchuk (2005).

[Figure]

Figure. Water flows between the Baltic Sea basins (km$^3$ year$^{-1}$) averaged over 1970-1999. External water inputs are the sum of net freshwater supply from river runoff and from atmosphere.

I would argue that the sub-basin boundaries in the model of Gustafsson etc also (2012) is not arbitrary chosen. As far as possible sub-basin boundaries of this model is chosen according to dynamical constraints such as sills or fronts that can be parametrizised. A discussion of the implications of the high-resolution sink/source fields for our understanding of major processes would have been quite interesting. What does the spatial distribution of e.g. net sedimentation or denitrification imply? What are the pathways for organic matter? I am not sure how far you can take this given methodological limitations, but it could be nice here with a few things and not only referring to other model simulations.

**Response:** We clarify the boundaries description in Gustafsson et al. (2012). And we will consider the possibility to add some discussion about the high-resolution sink/source fields in the revised manuscript.

---

## Author Response (AR2)

Revised manuscript has been examined by one of the original reviewers. reviewer still finds two important points that need to be addressed:
1) unknown and variable magnitude of inherent non-conservativeness, which can compromise both estimates of internal sources/sinks and budgets.
 2) nutrient budgets must be re-evaluated and presented in a realistic way, including appropriate revision of Discussion with comparison to other studies.

REPLY: Thanks for your highlighting two points from reviewer. We have addressed all the comments from the Dr. Savchuk Oleg. Please find our revised manuscript.

Further reviewer's comments to revised manuscript by Y. Liu, H.E.M. Meier and K. Eilola "Nutrient transport in the Baltic Sea - results from a 30-year physical-biogeochemical reanalysis" submitted to "Biogeosciences"

The manuscript has been substantially improved in many aspects, from clarification of procedures and algorithms to its language. However, besides of possible further stylistic cosmetics, there is still a couple of important obscurities left, which need to be either clarified or entirely removed in order to increase the paper's credibility because retaining clearly questionable issues reduces confidence in more plausible (reliable) results.

1. General comments and suggestions

1.1. As the major objectives of the data assimilation, the revised Introduction lists: a) the reconstruction of the water quality with high resolution, especially in under-sampled periods and areas, b) the estimation of nutrient transports from more realistic nutrient fields, c) the implementation of reconstructed fields as initial and boundary conditions. In addition to these, I would also explicitly stress the capability of dynamical estimating of the water nutrient pools, especially their long-term developments, as indicators of the trophic state (see 2.2 below).

REPLY: We add in the introduction: "A good reanalysis of biogeochemical state variables can dynamically describe indicators of eutrophication such as the long-term development of water nutrient pools."

We add a line to section 6.3 after "The results of the reanalysis can be used to estimate the water quality and ecological state with high spatial and temporal resolution in regions and during periods when no measurements are available."
"This supports improved assessments e.g. of eutrophication status indicators."

We added to section 5.6 (see also reply to 2.2 below): "The 3D nutrient pools constructed by data assimilation methods offer an opportunity to evaluate with improved estimates the changes in Baltic Sea eutrophication. …

1.2. However, I still miss the explicit indication of limitations and applicability of this approach already in Introduction and even, perhaps, in Abstract. In your response to me you indicated: "In the introduction section we further clarify the already listed limitations of data assimilation with respect to estimating nutrient budgets and we rewrite the objectives of this study." However, the first mention: "As a reanalysis can never be dynamical consistent and does not preserve mass, momentum and energy…" (Lines 100 – 101) of non-conservativeness seems appeared here out of nowhere. Consider, please,
a) re-formulation as "As a reanalysis does not CONSERVE mass, momentum and energy and, (var. - therefore, consequently, thus) can never be dynamicalLY consistent and …" and b) moving to- (or making similar statement) somewhere between lines 68-72.

REPLY: We have changed the corresponding text and move them to the lines 68-72.

1.3. Most important among such limitations is an inherent non-conservativeness of the approach. As you correctly admitted in several places of your responses: "The potential impact from artificial sources or sinks due to data assimilation is of course also included in the reanalysis results. Because of the unknown impact from this "process" it is better to avoid detailed discussions …especially about the changes in the nutrient pools… about the net sources and sinks." First of all, why "potential" when it is a real artificial change of simulated masses of nutrients (see also lines 423-425)? Unfortunately, you have also neglected to estimate its possible magnitude as a difference between 3D pools estimated before and after assimilation, e.g. simultaneously with RMSD (lines 218-219). I understand that it might be too laborious to systematically calculate such differences over the entire integration period but then it could be made as a few cases, perhaps, at the very first "act of assimilation" and then a couple of times more, just to show a magnitude. Such estimates are important because I can hardly see a justification of your statement at lines 427-429 in Fig. 3, where RMSDs do not show really significant differences between the beginning and the rest of REAN experiment.

REPLY: We agree that artificial sources or sinks affect the reanalysis results.
The figure below illustrate the magnitude of RMSD difference before and after data assimilation in REANA. The changed magnitude of RMSD by data assimilation is small, which support our text in the discussion --"data assimilation has corrected the mismatch between model state and observation to an "optimal" level during an initial adjustment period. After the adjustment period, the mismatch between model and observation becomes small and the successive adjustment due to data assimilation also becomes small."

[Figure]

Figure 1. The monthly averaged RMSD before and after the data assimilation and their residual (After-Before) in REANA during the period 1970-1971.

1.4. Furthermore, I now think that instead of "avoiding the discussion" about erroneous sources and sinks, their usage and analyses should be excluded entirely. Consider, please, for instance, your estimates of internal sources and thinks as a difference between integrated inflows and outflows of nutrients into every grid cell, where water transports is calculated as a product of the concentrations and the current velocity vectors in the considered and surrounding cells (Eq. 5). Without data assimilation these concentrations are determined by the explicitly parametrized pelagic and sediment biogeochemical processes and, thus, the resulted differences can justifiably be interpreted as sources and sinks of the model variables. In the case of data assimilation, these concentrations are artificially altered by unknown values, unknowingly variable both in time and space. These dynamical errors are then uncontrollably propagated over the simulation domain. As you correctly noted at lines 458-460: "…usually small artificial sources and sinks from the data assimilation are becoming as important as physically motivated sources and sinks when sums of fluxes are compared." Therefore, your estimates are not strictly comparable to those in Eilola et al. (2012) and lead to errors if being implemented in budgeting. As you say: "In this study, we focus mainly on nutrient transports derived from the reanalysis" (Line 207) and I advise to stay strictly with that.

REPLY:  We agree with your comments and hope that the text of the revised manuscript explains the focus of our study better. However, we keep the revised Figures 10 and 11 in the revised manuscript because we believe that the comparison of transports between sub-basins between our and previous studies is still very interesting. As the calculation of transports across sections follows conservation principles our estimates of nutrient exchange between sub-basins are reliable for the considered time period. The revised calculation of the integrated sources/sinks within the sub-basins includes of course the artificial sources/sinks caused by the data assimilation method and makes an interpretation difficult. However, as our numbers of sources/sinks are similar to those of previous studies we conclude that our overall budgets are reliable.

1.4. Nutrient budgets.

1.4.1. Definitions and algorithms. It is still not clear enough, how the "burial" terms in Figs. 9-10 were calculated. Terminologically, is it a proxy of real sediment burial (i.e. permanent removal from the model domain) or a sediment retention/accumulation, especially at a long-term (1970-1999) budgeting scale? Or even simpler – is it just integral sink or nutrient removal, accounting also for denitrification? Is it actually a difference between nitrogen fixation and denitrification that is accounted here for? Do you consider these "burial" values as "the sediment budgets" (line 99)? If it is the integration of sources/sinks described in Section 5.5 then the critique from 1.3 above apply. If "The sediment sinks (burial) are calculated from the difference between the net deposition of nutrients to the sediments and the release of nutrients from the sediments" (lines 227-228) then it should be better explained a) what does "net deposition" mean – was it explicitly calculated as the difference between total sedimentation of plankton+detritus+ resuspended sediments minus uplifted sediments and b) how the sediment release of nutrients was computed, why it is not presented and analysed, which could be more relevant and interesting than the entire Section 5.5? Please, consider carefully and explain more precisely.

REPLY: we recalculated the nitrogen and phosphorus sink in the Figs10-11 in revised manuscript. We describe this in section 4:  "The sinks of the nutrient budgets are calculated from the supplies from land/atmosphere, import/export from other basins and the changes in pelagic nutrient pools during the period (sink=supply+import-export-pool change)."

We removed all discussions about tendencies.

1.4.2.   There   are   also   internally   contradicting   statements   and   unfulfilled   intentions:

Lines 100-102 – "As a reanalysis … does not preserve mass… the calculated budgets are compared to … other studies … as consistency check". OS: – if only within an order-of-magnitude, because of different approaches and compared periods.

REPLY: We rephrased in section 6.3." Further, nutrient transports across selected cross-sections or between vertical layers are calculated from the reanalysis with high resolution and improved accuracy. However, one cannot expect that budgets calculated from the summation of internal fluxes from model results with data assimilation are more accurate because usually small artificial sources and sinks from the data assimilation are becoming as important as physically motivated sources and sinks when sums of fluxes are compared. Hence, we calculated (section 5.6) budgets only from inputs and exports and changes in the water pools of nutrients with the aim to compare the reanalysis results with other studies using only observations. It is perhaps not possible to claim that our budgets are more accurate than budgets that are derived from observations only, despite the higher temporal and spatial resolution in model outputs. However, the advantage of the reanalysis is that measurements are extrapolated in space and time based upon physical principles of the model."

Lines 459-464 – "…one cannot expect that budgets calculated … with data assimilation are more accurate… Hence, we calculated budgets with the aim … to estimate the magnitude of artificial sources and sinks by comparing our results with other studies…" OS: Such estimate was not made.

REPLY: See above.

1.4.3. Most likely, the above inconsistencies are the reason why the presented budgets as a whole are highly unrealistic. The budget's terms are presented as average for 30 years including "tendencies" from Table 1. Then the starting and final total stocks (TS) can formally be calculated as follows: TSinit = TSaverage – dTS*15 and TSfinal = TSaverage + dTS*15. For the Gulf of Finland such estimates give in kT: Pinit = 29.9 – 3.7*15 = -25.6, Pfinal = 29.9 + 3.7*15 = 85.4, Ninit = 127 – 16*15 = -113, Nfinal = 127 + 16*15+ 367; for the Kattegat: Pinit = -29.3, Pfinal = 51.7, Ninit = -378, Nfinal = 522, etc.

At the same time, both the total pools and estimated nutrient exports and imports across the basin boundaries looks realistic enough to serve as another solid source of information for other studies and conclusions, if not being compromised by the doubts in "burial" terms estimated with artificial effects of assimilation.

REPLY: we re-estimated the nitrogen and phosphorus sink of the Baltic (please refer to the reply 1.4.1).

1.5. Algorithm of the data assimilation. Now it looks more comprehensive and save the reader from looking into similar papers. Unfortunately, it still too formal and, to my mind, is not narratively explanative enough for fellow marine biologists and biogeochemists. Perhaps, that is why it leaves (or arises) some additional questions, such as, for instance, about construction of BEC from simulation for 1962-1968, when erroneous effects of the poorly prescribed sediment (especially, P) nutrients should be even larger.

REPLY: The analysis quality of an ensemble data assimilation method is related to the BEC constructed by samples ensemble anomalies. We do not know the "truth" BEC. Typically we use a "good " sample ensemble for constructing BEC. These samples of EnOI are usually obtained from a historical simulation without data assimilation. The ensemble samples are not integrated in the simulation process. As a result, EnOI is a 3D data assimilation method and the model errors are assumed not to be integrated in simulation process. Actually the analysis increment is a combination of sample ensemble anomalies. In general the variance magnitude of the perturbation of the sample ensemble from historical simulation is not close to the one at the analysis time. Therefore EnOI use a inflate factor (alpha in Equtaion 3, 0<alpha<=1.0) to adjust the ensemble samples perturbation's variance. Our reanalysis simulation stated from 1970. We would like to construct the samples from the model snapshots before starting our reanalysis. Further, there are also many observations available and the model results have been validated in the period 1964-1968. Therefore, in this study, we select the samples from free run during the period 1964-1968. In order to keep the "independence" of sample, the snapshots during the period 1964-1968 have been stored every three days. The assimilation scheme in this study is same as that used in Liu et al. (2014). Other sample ensembles could be tested but this is beyond the scope of this study.

When observations become available at the assimilation time, the 'optimal' state variables are produced by Equtaion (1). These 'optimal' state variables are used as 'new' initial condition for the 'next' simulation cycle.

An overall flow diagram below explains the method qualitatively neglected all details.

[Figure]

Figure 3. Work flow diagram of the "weakly" coupled data assimilation.

2. Specific comments and suggestions.

2.1. Consider, please, re-estimation of nutrient budgets using only external nutrient inputs and nutrient flows across basin's boundaries. The sinks/sources can then be found as the rest terms assuming either steady state or accounting also for the differences between initial and final pools (see 2.2) below. Then, the analysis of budgets should be revised accordingly both in Results and Discussion.

REPLY: we have recalculated the sink/sources according to your advices in the revised manuscript.

2.2. The potential of the integrated nutrient pools estimation from more realistic (?) 3D nutrient fields reconstructed by data assimilation into 3D coupled physical-biogeochemical could be exploited much more for description of eutrophication. For that as well as for the budgeting, it is not even necessary to calculate annual dynamics but just, say, 5 (10?) year averages in the beginning and end of simulation.

REPLY: We added to section 5.6: "The 3D nutrient pools constructed by data assimilation methods offer an opportunity to evaluate with improved estimates the changes in Baltic Sea eutrophication. As a example, an investigation of the trophic state from changes in five year average nutrient pools in REANA show that the total nitrogen pool in the Baltic proper increased from 657 kton to 1045 kton from the period 1971-1975 to 1995-1999 while the total phosphorus pool decreased from 469 kton to 448 kton between the same periods. Hence, nitrogen increased by about 59% while phosphorus decreased by about 4%. Similarly the pool of DIN in REANA increased by 80% while DIP decreased by 6%. The corresponding numbers obtained from BED showed an increase of 100% for DIN and an increase of 7% of DIP. The results indicate large increases of nitrogen pools in the Baltic proper during the investigated period but only relatively small changes of phosphorus pools.

Subsequent periodic assessments can be used to reveal future eutrophication changes. While estimating the trophic state, it should be noticed that the change in trophic state depends on the chosen time periods. For example, from the year 1971 to 1999 the total phosphorus (TP) in the Baltic proper increased with 2.7 kton yr$^{-1}$. This differs from the decrease seen from the five year average change discussed above. The reason is the impact from short term fluctuations of nutrient content (Figure 4) that may be larger than the long-term changes."

2.3. In addition to running RMSDs, a comparison of such average pools between REAN and FREE simulations (perhaps, also with BED) as the measure of improvement could be interesting too,

REPLY: We added to section 5.1: "The annual averaged pelagic pool of simulated DIN and DIP were also compared to the corresponding pools estimated from the BED data in Baltic proper. The maximal annual differences of DIN and DIP compared to the BED, have been reduced by 57.5% and 72.3%, respectively, from 400 kton and 650 kton in FREE (not shown) to 170 kton and 180 kton in REANA. The remaining differences between REANA and BED may be explained by the methods of integration that differ. BED estimates are based on a limited amount of observations while the model results are based on a large number of grid points with dynamically varying state variables."

[Figure]

Figure 4. Annual mean integrated pools (in kton) of pelagic dissolved inorganic nitrogen (DIN) and phosphorus (DIP) in the Baltic proper calculated from REANA and from observations with DAS.

2.4. End of Section 3. A few sentences are needed here to explain what does indication "in BED" in Sections 5.2-3 mean and how the long-term average March fields in 5.3 are reconstructed.

REPLY: We have added the text in the Section 3. "The simulated spatial variations of the late winter surface layer nutrient concentrations are compared with the spatial variations reconstructed from BED with the Data Assimilation System (DAS) by Sokolov et al. (1997). Due to insufficient historical data coverage the average March fields were computed for time period (1995–2005) from over 3600 oceanographic stations found in BED. Also nutrient pools of dissolved inorganic nitrogen (DIN) and phosphorus (DIP) calculated with DAS (see Savchuk, 2010) are compared with the results of this study. See Eilola et al. (2011) for more details about the data handling by DAS."

And in Section 5.3 we added: "The average March concentrations (1970-1999) of DIP and DIN in the upper layers (0–10m), as well as their ratio (DIN:DIP), were calculated (Fig. 6). Due to insufficient historical data the corresponding BED maps describe the averages for the period 1995-2005 as a basis for model to data comparison."

2.5. Lines 220-224. As follows from Eq. 5, the product of velocity vector "Vk" and concentration "Ck", both taken at the location "k", represents just transports (or outflowing nutrient mass) at this location and not "net transports", which word better to keep for real differences between integral inflows and outflows, like you use for derivation of (semi-artificial) internal sources and sinks in Section 5.5. as well as for more plausible really net transports between basins and along cross-sections in Section 5.6. Please, check carefully the usage of "net transports" and correct accordingly, for instance, at line 295.

REPLY: Throughout the entire manuscript we changed the definitions. We differentiate now between "vertically integrated transports" and "net transports". The latter is the difference between export and import of volume or nutrients integrated across the sections in sub-basins.

2.6. Line 273-274 – an improved model description of vertical transports of nutrients in the layers above the halocline." Since Fig. 4 says nothing about water movements, explain, please, the reason(s) for a statement about improvement of transports – is it due to realistic vertical gradients or something else? Where is this claimed analysis of vertical transports?

REPLY: We have improved the modeling stratification which affects the vertical transports of nutrients, especially the water depths around the thermocline. The vertical distributions of nutrients have been improved above the halocline (Figure 5). For clarify, we changed the text to "an improved model description of vertical DISTRIBUTIONS of nutrients in the layers above the halocline"

2.7. Consider, please, making a separate Section starting from line 372

REPLY: we have done it in revised manuscript.

3. Minor things, technical corrections and language cosmetics

REPLY: Thanks for your comments. We have changed these texts in revised version of manuscript.
Title – I feel, "Nutrient transportS…" would sound better

Line 97 - "This information can't be obtained from neither observations alone or…" The grammar construction should be either "…CAN NOT … EITHER… OR…" or " CAN… NEITHER… NOR…", I would recommend the former

Line 100 - "…dynamicalLY consistent.." - it is adverb

Line 101 – replace, please, "preserve" by "conserve" because we are dealing here with the laws of conservation of mass and energy…

Section 5.1 – Please, restore reference to Fig. 3

Line 408 – "…nutrients are not detailed enough known." Please, rephrase less clumsy: "are poorly known", "are insufficiently known", "are known not detailed enough", are known not in enough detail"

Line 752 – Ref. to Terruzi et al (2014) should be to pp. 200-217

Line 817 –"…Northwestern Gotland Basin, Bothnian Sea…" – Gulf of Finland is missing in this listing.

[revised manuscript text omitted]

The assimilation of data into coupled physical-biogeochemical models is confronted by various theoretical and practical challenges. For example, the response of the three-dimensional biogeochemical model to external forcing caused by the physical model is highly non-linear. Further, it is difficult to use the biological observational information to reduce biases in the simulation of ocean physics which has an impact on modeled biogeochemistry (Beal et al., 2010). Besides, data assimilation as used in this study does not conserve mass, momentum and energy. Therefore, a reanalysis with data assimilation can never be dynamically fully consistent.

Nevertheless, the use of data assimilation complementing ecosystem modeling efforts has gained widespread attention (e.g. Hoteit et al., 2003; Allen et al., 2003; Natvik and Evensen, 2003; Hoteit et al., 2005; Triantafyllou et al., 2007; While et al., 2012; Triantafyllou et al., 2013; Teruzzi et al., 2014). Data assimilation into ecosystem models has focused both on parameter optimization and on state and flux estimations (Gregg et al, 2009). A comprehensive review of biological data assimilation experiments can be found in Gregg et al. (2009).

In the Baltic Sea, the biogeochemical data assimilation has started to become a research focus. For example, Liu et al. (2014) used the Ensemble Optimal Interpolation (EnOI) method to improve the multi-annual, high-resolution modelling of biogeochemical dynamics in the Baltic Sea. Fu (2016) analyzed the response of a coupled physical-biogeochemical model to the improved hydrodynamics in the Baltic Sea. Recently, several data assimilation studies have focused on the historical reanalysis of salinity and temperature in the Baltic Sea (e.g. Fu et al., 2012; Liu et al., 2013; 2014). Reanalysis has helped enormously in making the historical record of observed ocean parameters more homogeneous and useful for many purposes. For instance, ocean reanalysis data have been applied in research on ocean climate variability as well as on the variability of biogeochemistry and ecosystems (e.g. Bengtsson et al., 2004; Carton et al., 2005; Friedrichs et al., 2006). Ocean reanalysis can also be used for the validation of a wide range of model results (e.g. Fontana et al., 2013). For instance, the ocean mean state and circulation can be calculated from reanalysis results to evaluate regional climate ocean models (e.g. Meier et al., 2012). Moreover, reanalysis in the ocean is beneficial to the identification and correction of deficiencies in the observational records, as well as filling the gaps in observations. Regional and local model studies may use reanalysis results as initial and boundary conditions. A good reanalysis of biogeochemical state variables can dynamically describe indicators of eutrophication such as the long-term development of water nutrient pools.

[revised manuscript text omitted]

   The simulated spatial variations of the late winter surface layer nutrient concentrations are compared with the spatial variations reconstructed from BED with the Data Assimilation System (DAS) by Sokolov et al. (1997). Due to insufficient historical data coverage the average March fields were computed for time period (1995–2005) from over 3600 oceanographic stations found in BED. Also nutrient pools of dissolved inorganic nitrogen (DIN) and phosphorus (DIP) calculated with DAS (see Savchuk, 2010) are compared with the results of this study. See Eilola et al. (2011) for more details about the data handling by DAS.

    **4 Methodology and Experimental Setup**

Here we briefly describe the configuration of the data assimilation system of this study. We focus on the state estimation via EnOI. The distribution of stochastic errors are assumed to be Gaussian and non-biased. EnOI

estimates an 'optimal' oceanic state at a given time using observations, the numerical model and assumptions on their respective bias distribution. The relationship between them can be expressed as following:

$$\psi^a = \psi^f + \mathbf{K}(d - H\psi^f) \qquad (1),$$

$$\mathbf{K} = \mathbf{P}^f H^{\mathrm{T}} (H\mathbf{P}^f H^{\mathrm{T}} + (N-1)\mathbf{R})^{-1} \qquad (2).$$

Where $d \in \Re^m$ is the vector of observations with $m$ being the number of observations. $\psi \in \Re^n$ is the $n$

dimensional model state vector which includes the sea level anomaly, temperature, salinity, oxygen, phosphate, ammonium and nitrate. The superscripts $a$ and $f$ refer to "analysis" and "forecast", respectively. $\mathbf{K} \in \Re^{n \times m}$ is the Kalman gain matrix and $H$ is an operator that maps the model state onto the observation space–often $H$ is linear interpolation. $d - H\psi^f \in \Re^m$ is the innovation which is calculated in the observation space. $\mathbf{R} \in \Re^{m \times m}$ is the observation error covariance. $N$ is the number of the ensemble samples. EnOI computes the Background

Error Covariance (BEC) matrix $\mathbf{P} \in \Re^{n \times n}$, which determines how to spread out information from observations in space and between variables, by the ensemble perturbation matrix $\mathbf{A}' = \mathbf{A} - \overline{\mathbf{A}}$ as follows:

$$\mathbf{P} = \frac{\alpha}{N-1} \mathbf{A}'(\mathbf{A}')^{\mathrm{T}} \qquad (3).$$

Here $\mathbf{A} = (\psi_1, \psi_2 ..., \psi_N) \in \Re^{n \times N}$ is the sample ensemble and $\overline{\mathbf{A}} = \frac{1}{N} \sum_{i=1}^{N} \psi_i$ is the sample ensemble mean. The subscript T denotes the transpose of a matrix and the scaling factor $\alpha \in (0,1]$ is introduced to tune the variance of the sample ensemble perturbations to a realistic level in order to capture the variability of model parameters like temperature and dissolved oxygen, which is dominated by misplacement of mesoscale features and which varies in location and intensity seasonally. Therefore, we hypothesize that the background errors are proportional to the model variability on intra-seasonal time scales. We selected the samples from model results of a hindcast simulation without data assimilation from one and a half month before and after the calendar date of the assimilation time during the period 1964–1968 (Liu et al., 2013). The snapshots during the period 1964-1968 have been stored every three days. From every year during the selected period 1964–1968 20 snapshots have been selected. Hence, a total of $N = 100$ model samples are adopted to obtain a quasi-stationary BEC matrix. The analysis by EnOI rely on the sample ensemble because the analysis increment is a linear combination of sample ensemble anomalies. Other, more "sophisticated" sample ensembles could be tested but this is beyond the scope of this study. An adaptive scaling factor was calculated to adapt to the instantaneous forecast error variance before each local analysis (Liu et al., 2013; 2014). Further, localization is used to remove unrealistic long-range correlation with a quasi-Gaussian function and a uniform horizontal correlation scale of 70 km. As a result, the quality of fields obtained by data assimilation is determined by the coverage and quality of observations (She et al., 2007). Moreover, the assimilation frequency or window is another factor to affect the assimilation fields. They are directly related to how many observations are entering the assimilation cycling and how often the model initial condition is adjusted by data assimilation (Liu et al., 2013). Here, we select an assimilation window of three days and the assimilation frequency is once every seven days in the reanalysis experiment. It means that all the observations in three days before and after the assimilation time are selected to yield the "new" initial condition for the following simulation during the current assimilation cycle. When observations become available at a certain time, the 'optimal' state variables are calculated by Equation 1, which are used as new initial conditions for the next simulation cycle.

[revised manuscript text omitted]

Further, the annual averaged pelagic pools of simulated DIN and DIP in the Baltic proper are compared to the corresponding pools estimated from the BED data in the Baltic proper (Fig. 4). The maximum annual differences of DIN and DIP pools compared to BED, have been reduced by 57.5% and 72.3%, respectively, from 400 kton and 650 kton in FREE (not shown) to 170 kton and 180 kton in REANA. The remaining differences between REANA and BED may be explained by the methods of integration that differ. BED estimates are based on a limited amount of observations while the model results are based on a large number of grid points with dynamically varying state variables. A similar result for hypoxic area was found by Väli et al. (2013). They showed with the help of model results which were sampled at the same times and locations of the observations that the applied interpolation algorithm underestimated hypoxic area by about 40%.

**5.2 Mean seasonal cycle of nutrients**

The long-term average seasonal cycles of temperature and inorganic nutrients at monitoring station BY15 at Gotland Deep (for the location, see Fig. 1) give a hint of how data assimilation improves simulated nutrient dynamics in the Baltic proper (Fig. 5). The surface layer temperature and stratification show rapid increase in April to May, with concurrent rapid decrease of nutrient concentrations due to primary production down to 50–60 m depths. The cooling and increased vertical mixing in autumn and winter reduce temperatures and bring nutrients from the deeper layers into the surface layers. RCO-SCOBI captures these variations. However, compared to BED, FREE has obvious biases, such as overestimated temperature stratification around 30–50 m depth from late winter to early spring, higher concentration of nutrients at the 50–60m depth, stronger vertical stratification of nutrient concentrations and less decrease of nutrients in the summer, especially below the thermocline, as well as also in the surface layers for phosphate. One reason for the biases is the vertical displacement of the halocline that is too shallow in RCO (e.g. Fig. 4 in Liu et al., 2014). The causes for the model bias in nutrient depletion below the summer thermocline are not known, but possible reasons are discussed by Eilola et al. (2011). These biases are significantly reduced in the reanalysis which provides an improved description of vertical distributions of nutrients in the layers above the halocline.

**5.3 Spatial variations of late winter nutrient concentrations**

The average March concentrations (1970-1999) of DIP and DIN in the upper layers (0–10m), as well as their ratio (DIN:DIP), were calculated (Fig. 6). Due to insufficient historical data the corresponding BED maps describe the averages for the period 1995–2005 as a basis for model to data comparison. In BED results 
[revised manuscript text omitted]
 (Figs. 10 and 11). Changes in pools are calculated as differences between 1971 and 1999 because the initial adjustment process due to the assimilation is taking place during the first year (1970) (not shown). The largest annual mean external phosphorus load occurs in the Baltic proper and amounts to 34.2 kton $yr^{-1}$ (Fig. 10). In addition, in the Baltic proper the largest annual mean phosphorus sink of 25.0 kton $yr^{-1}$ is also found. Whereas during the period 1971–1999 the phosphorus content in the Gulf of Baltic proper increased, we found decreasing phosphorus content in the Gulf of Finland, Bothnian Bay, Bothnian Sea and Danish Straits. Largest export and import of phosphorus between sub-basins are found for the exchange between the Baltic proper and the Gulf of Finland, which amount to 24.3 and 22.5 kton $yr^{-1}$, respectively. However, the largest net exchange (import minus export) appears between the Baltic proper and Bothnian Sea. It is also found that the Baltic proper exports more phosphorus to neighboring sub-basins than it imports, except for the Gulf of Riga. The annual mean net phosphorus exported from the Baltic proper into the Danish Straits, the Bothnian Sea, the Gulf of Finland and Gulf of Riga during the period 1971–1999 amounts to 1.7, 3.6, 1.8, and -0.6 kton $yr^{-1}$, respectively. The exchange of phosphorus between the Baltic proper and the Gulf of Riga is smallest relative to the other three neighboring sub-basins. Further, we found that the net transport, import and export of phosphorus into the Bothnian Bay are smallest relative to the other sub-basins.

Nitrogen transports between Baltic Sea sub-basins are different compared to phosphorus transports (Fig. 11). For example, the Baltic proper has larger nitrogen sinks than external sources. Further, during the period 1971–1999 the nitrogen content decreased in the Gulf of Riga and increased in the Bothnian Bay, respectively. In the Gulf of Finland and Danish Straits, the difference between external supply and internal sink of nitrogen is equal to the net transport into the Gulf of Finland and Danish Straits. The large sink of nitrogen in the Bothnian Bay is noteworthy. We also found relatively large net transports of nitrogen from the Gulf of Riga into the Baltic proper. This is mainly explained by the relatively high nitrate concentrations in the Gulf of Riga relative to other sub-basins.

The 3D nutrient pools constructed by data assimilation methods offer an opportunity to evaluate with improved estimates the changes in Baltic Sea eutrophication. As an example, an investigation of the trophic state from changes in five year average nutrient pools in REANA shows that the total nitrogen pool in the Baltic proper increased from 657 kton to 1045 kton from the period 1971-1975 to 1995-1999 while the total phosphorus pool decreased from 469 kton to 448 kton between the same periods. Hence, nitrogen increased by about 59% while phosphorus decreased by about 4%. Similarly the pool of DIN in REANA increased by 80% while DIP decreased by 6%. The corresponding numbers obtained from BED showed an increase of 100% for DIN and an increase of 7% of DIP. The results indicate large increases of nitrogen pools in the Baltic proper during the investigated period but only relatively small changes of phosphorus pools.

Subsequent periodic assessments can be used to reveal future eutrophication changes. While estimating the trophic state, it should be noticed that the change in trophic state depends on the chosen time periods. For example, from the year 1971 to 1999 the total phosphorus (TP) in the Baltic proper increased with 2.7 kton yr$^{-1}$. This result differs from the decrease seen from the five year average change discussed above. The reason is the impact from short term fluctuations of nutrient content (Fig. 4) that may be larger than the long-term changes.

**5.7 Baltic nutrient flows**

[revised manuscript text omitted]

Sokolov, A., Andrejev O., Wulff, F., and Rodriguez Medina, M.: The Data Assimilation System for Data
Analysis in the Baltic Sea. Systems Ecology Contributions, 3, Stockholm University, 1997, 66pp.

Stevens, D. P.: The open boundary conditions in the United Kingdom fine-resolution Antarctic model, J. Phys.
Oceanogr., 21, 1494–1499, 1991.

Teruzzi, A., Dobricic, S., Solidoro, C., and Cossarini, G.: A 3-D variational assimilation scheme in coupled
transport-biogeochemical models: Forecast of Mediterranean biogeochemical properties, J. Geophys. Res.
Oceans, 119, 200-217, 2014.

Triantafyllou, G., Korres, G., Hoteit, I., Petihakis, G., and Banks, A.C.: Assimilation of ocean colour data into a
Biogeochemical Flux Model of the Eastern Mediterranean Sea Ocean Sci., 3, 397–410, 2007.

Triantafyllou, G., Hoteit, I., Luo, X., Tsiaras, K., and Petihakis, G.: Assessing a robust ensemble-based Kalman filter for efficient ecosystem data assimilation of the Cretan Sea,  Journal of Marine Systems, 125, 90–100,
2013.

Väli, G., Meier, H.E.M., and Elken, J.: Simulated halocline variability in the Baltic Sea and its impact on hypoxia
during 1961-2007, J. Geophys. Res. Oceans, 118, 6982-7000, doi:10.1002/2013JC009192, 2013.

Voss, M., Emeis, K.-C., Hille, S., Neumann, T., and Dippner, J.W.: Nitrogen cycle of the Baltic Sea from an
isotopic perspective. Global Biogeochemical Cycles 19: GB3001. doi: 10.1029/2004GB002338, 2005.

While, J., Totterdell, I., and Martin, M.: Assimilation of $pCO_2$ data into a global coupled physical-
biogeochemical ocean model, J. Geophys. Res., 117, C03037, doi:10.1029/2010JC006815, 2012.

Wulff, F., Rahm, L., Larsson, P. (Eds.).: A systems analysis of the Baltic Sea: Ecological Studies, Vol. 148.
Springer, Berlin, 2001.

Wulff, F., and Stigebrandt, A.: A time-dependent budget model fro nutrients in the Baltic Sea. Global
Biogeochemical Cycles, 3(1), 63–78, 1989.

[Figure]

Figure 1. The bathymetry of the model (depth in m). The border locations of sub-basins of the Baltic Sea used in this study are shown by the black lines, and the BY15 station is shown by the white star. Names of the sub-basins are the Kattegat (KT), Danish Straits (DS), the Baltic proper (BP), the Gulf of Riga (GR), the Gulf of Finland (GF), the Bothnian Sea (BS), and the Bothnian Bay (BB).

[Figure]

Figure 2. Number of observed profiles in different sub-basins (upper panel) and annual number of profiles from
1970–1999 (bottom panel).

[Figure]

Figure 3. Monthly mean root mean square deviation (RMSD) between model results and observations for
oxygen, nitrate, phosphate and ammonium in FREE (red) and REANA (blue).

[Figure]

Figure 4. Annual mean integrated pools (in kton) of pelagic DIN and DIP in the Baltic proper calculated from REANA and from observations in BED.

[Figure]

Figure 5. The seasonal cycle of monthly average (1970–1999) temperature (°C), phosphate concentration (mmol m$^{-3}$), and nitrate concentration (mmol m$^{-3}$) at BY15 for FREE (row 1), REANA (row 2), and BED data (row 3), respectively.

[Figure]

Figure 6. Simulated monthly (March) mean (1970–1999) surface layer (0–10 m) concentrations of DIP (mmol m$^{-3}$) (left), DIN (mmol m$^{-3}$) (middle), and the corresponding DIN to DIP ratio (right) from FREE and REANA are shown in rows 1 and 2, respectively. The corresponding BED maps in row 3 are calculated from observations monitored during the period 1995-2005.

[Figure]

Figure 7. Annual mean DIP transports and the corresponding DIP persistency, DIN and OrgP transports for REANA averaged for the period 1970–1999. The black solid lines with arrows show the streamlines and direction of transports. The magnitude of transports (kton km$^{-1}$ yr$^{-1}$) and the persistency (%) are shown by the background color. The corresponding values are shown in the colored bars.

[Figure]

Figure 8. Spatial distributions of annual mean import of DIP, OrgP, DIN and OrgN averaged for the period 1970–1999. The magnitude of import and its corresponding value (kton km$^{-2}$ yr$^{-1}$) are shown by the background color and color bar, respectively. Green colors denote positive values (import), and yellow to red colors denote negative values (export). The black and blue lines show 30 and 100 m depth contours of the model, respectively.

[Figure]

Figure 9. Annual mean, accumulated net imports (black lines) and imports of OrgP, DIP and DIN (color bars) to regions with the same depth in the Baltic proper averaged for the period 1970–1999.

[Figure]

Figure 10. Annual mean total phosphorus budgets of the Baltic Sea averaged for the period 1971–1999. The average total amounts are in kton, and transport flows and sink/source fluxes (external nutrient inputs/sink) are in kton yr$^{-1}$. External nutrient inputs from atmosphere and land are combined.

[Figure]

Figure 11. The same as Figure 9, but for nitrogen.

[Figure]

Figure 12. Annual mean fluxes of nitrogen (in kton yr$^{-1}$) and phosphorus (in kton yr$^{-1}$) as a function of the cross sections along transects following the latitude and longitude in the Baltic sub-basins. Northward and eastward fluxes are, by definition, positive and called inflows. Southward and westward flows are called outflows. Net flow is the difference between in- and outflows. Here, AR, BH, GO, NW, GF, BS, and BB represent the Arkona

Sea, Bornholm Sea, Eastern Gotland Basin, Northwestern Gotland Basin, Gulf of Finland, Bothnian Sea and

Bothnian Bay, respectively. Transect A summarizes fluxes from the southern Baltic proper to the Bothnian Bay.

Transect B describes the Baltic Sea entrance area from the Arkona Basin to the Bornholm Basin, and transect C

summarizes fluxes in the Gulf of Finland (see Fig. 1).